# A statistical resolution measure of fluorescence microscopy with finite photons

Yilun Li[1] & Fang Huang [1,2,3] ✉

First discovered by Ernest Abbe in 1873, the resolution limit of a far-field microscope is considered determined by the numerical aperture and wavelength of light, approximately $\frac{\lambda}{2NA}$. With the advent of modern fluorescence microscopy and nanoscopy methods over the last century, this definition is insufficient to fully describe a microscope's resolving power. To determine the practical resolution limit of a fluorescence microscope, photon noise remains one essential factor yet to be incorporated in a statistics-based theoretical framework. We proposed an information density measure quantifying the theoretical resolving power of a fluorescence microscope in the condition of finite photons. The developed approach not only allows us to quantify the practical resolution limit of various fluorescence and super-resolution microscopy modalities but also offers the potential to predict the achievable resolution of a microscopy design under different photon levels.

In 1873, Abbe published his work stating that microscopy resolution solely depends on the numerical aperture and wavelength of light[1,2], a statement later verified theoretically[3,4]. This limit generally suffices for traditional microscopes, which collect transmission, reflection, or scattered light as signals[5]. The signal-to-noise ratio (SNR) can be optimized by adjusting the illumination power. However, in fluorescence microscopy, photons—the sole source of molecular information generated by individual fluorescent probes—are limited due to the photobleaching and photochemical environment of the fluorophores[6,7].

The discrete nature of light results in inherent photon counting noise, which follows a Poisson distribution. Consequently, the SNR diminishes as the number of detected photons decreases. This reduction in SNR at low photon levels complicates the distinction between actual structural differences and random noise fluctuations, thereby hindering the ability of microscopy techniques to reach their theoretical resolution limits[8–17].

In search of a practical resolution limit, researchers have attempted to create a measure based on the channel capacity from Shannon's information theorem[18], which takes noise into consideration[19–22]. The general concept is that a microscope's channel capacity stays constant and can be calculated from noise level and system configuration. Although these works provided the theoretical relationship between the channel capacity of the microscope and SNR, numerical aperture, etc., achieving high resolution requires a specific information encoding-decoding scheme. Furthermore, this channel capacity approach assumed that noise is (frequency) band-limited, a condition not met by photon noise. Another noise-considered resolution measure was proposed by ref. [16], who calculated the estimation precision of the distance between two closely spaced point objects. While this measure is well-suited for single molecule localization-based microscopy, for general microscopy modalities, the strong assumption of two separated emitters makes it challenging to converge with Abbe's conventional limit.

Other approaches to quantifying the effect of noise on resolution involve evaluating the noisy images directly[8,9,13,14,15,17]. For instance, by quantifying contrast, Stelzer[8] discussed noise's influence on practical resolution in confocal and wide-field systems, which represents one of the first demonstrations of the critical difference between practical resolvability in the presence of noise and Abbe's resolution limit. Another example used Fourier ring correlation[13,14,15], which quantifies practical resolution by evaluating the correlation between the spatial frequencies from two independently obtained images of the same structure[14,15]. Nonetheless, these empirical methods rely on existing microscopy data and well-controlled specimens. To date, a theoretical resolution measure that incorporates photon statistics is yet to be established.

[1]Weldon School of Biomedical Engineering, Purdue University, West Lafayette, IN, USA. [2]Purdue Institute for Integrative Neuroscience, Purdue University, West Lafayette, IN, USA. [3]Purdue Institute of Inflammation, Immunology and Infectious Disease, Purdue University, West Lafayette, IN, USA. ✉e-mail: fanghuang@purdue.edu

Here, we propose a theoretical measure for quantifying the resolving power of microscopes, accounting for numerical aperture, emission wavelength, and photon statistics. Our approach considers the Fisher information of a sinusoidal grating's phase estimation per area, defined as information density, to measure the imaging system's resolving power. Based on an adjustable criterion of the information density threshold, we define an information-based resolution (IbR). This measure is applied to evaluate and distinguish the significant practical resolution differences across various conventional and super-resolution imaging modalities, including wide-field microscopy, con-focal microscopy[23,24], two-beam structured illumination microscopy (SIM)[25,26], and image scanning microscopy (ISM)[27,28]. We expect IbR to be a useful measure in estimating the noise-considered resolution to guide and validate the design of newly developed or proposed imaging modalities.

## Results

### Model

To examine the resolution of an imaging system, we established theoretical models for planar and volumetric target specimen cases. In the planar case, we simulated a 2D single tone sinusoidal grating as our target object without background, meaning all photons were emitted by the sinusoidal grating object. In the volumetric case, we simulated a 2D single-tone sinusoidal pattern immersed in the focal plane located in the middle section of a 30 μm specimen volume. Background fluorescence was simulated using Sandison's and Gan's model[12,29], generated by a uniform distribution of fluorophores within the 3D volume. Background calculation in each imaging modality was the overlay from each axial section of the volume specimen (Methods). All imaging conditions were assumed to be fluorescence microscopes of symmetric systems.

We expressed the 2D sinusoidal grating object as

$$obj(x, y) = U_{ave}[1 + \sin(2\pi klx + \phi)]. \tag{1}$$

Here $(x, y)$ are pixel indices in the two-dimensional space. $U_{ave}$ denotes the average photon count per pixel, $l$ is the pixel length, $k$ is the spatial frequency of the sinusoidal grating along the x-axis, and $\phi$ is the initial phase of the sinusoidal grating.

The definition of Fisher information[30,31] is:

$$I(\phi) = E\left[\left(\frac{\partial L(\phi)}{\partial \phi}\right)\left(\frac{\partial L(\phi)}{\partial \phi}\right)^T\right], \tag{2}$$

where $\phi$ represents the parameters to be estimated, and $L(\phi)$ is the log-likelihood function of $\phi$. $E[\cdot]$ denotes the expectation taken over all possible outcomes (all possible noisy pixel values). Assuming the detected photons from the above object follow a Poisson distribution due to photon counting noise, we calculated the Fisher information as[32,33]

$$I(\phi) = \sum_i \frac{1}{u_i}\left(\frac{\partial u_i}{\partial \phi}\right)^2, \tag{3}$$

where $\boldsymbol{u} = [u_1, \ldots, u_i, \ldots, u_N]$ symbolizes the array of the expected photon counts from *obj* through different imaging modalities, and represents the expected photons of individual pixels. For wide-field and confocal systems, $\boldsymbol{u}$ corresponds to the ideal image (Methods). For conventional SIM, $\boldsymbol{u}$ represents nine frames of ideal images with different structured illumination patterns (Methods). For ISM, $\boldsymbol{u}$ consists of images corresponding to emission patterns recorded at different scanning positions (Methods).

Considering Fisher information $I(\phi)$ is additive, imaging the same object over a larger field of view (FOV) would result in a higher Fisher information, provided that the photon flux per area remains constant (Supplementary Fig. 1). We calculated the Fisher information per square microns, namely information density $I_d$. This information density $I_d$ measure has the unit of rad⁻²·μm⁻². We would like to note that $I_d$ was calculated in a deterministic manner based on Fisher information and optical theory, thus the information density can be considered as a theoretical function of the system and sample parameters. The visualizations, simulated as Poisson-noised images under various conditions for enhanced visual examination, were not employed in the calculation of Fisher information (Fig. 1).

To define a resolution limit, one needs a criterion for 'resolved.' For example, Rayleigh's criterion of $\frac{0.61\lambda}{NA}$ was defined as the distance of two points where their point spread function (PSF) first minima reach each other's center[34]. Sparrow et al., defined two points as resolvable if the mid-point of their joint intensity profile shows a minimum[35]. In search of a noise-considered resolution criterion, we set the threshold of information density $I_d$ at 10 rad⁻²·μm⁻² to serve as a resolving criterion in this work. This threshold was chosen so that one can no longer distinguish a sine pattern from cosine in a unit cycle of the target (Supplementary Fig. 2, Supplementary Note 1). Subsequently, IbR was defined as the reciprocal of the highest frequency that the imaging modality could resolve, given the above criterion (colored dashed line in Fig. 2a).

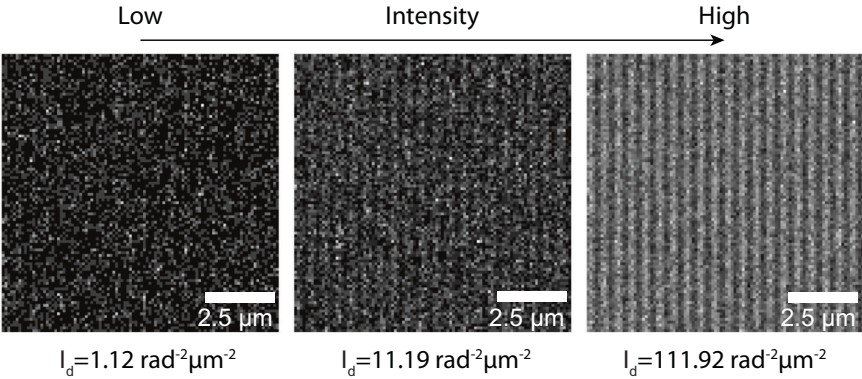

I_d=1.12 rad⁻²μm⁻²        I_d=11.19 rad⁻²μm⁻²        I_d=111.92 rad⁻²μm⁻²

**Fig. 1 | Visualization examples of noise's influence on the resolvability of a single-tone sinusoidal grating object.** The resolving power of this wide-field system is represented by the information density $I_d$ (unit: rad⁻²·μm⁻²). Noisy images were simulated with different intensities in this wide-field system at an object frequency of $\frac{1.6NA}{\lambda}$, with photon densities of 500 photons/μm² (left), 5000 photons/μm² (center), and 50,000 photons/μm² (right). $I_d$ is the Fisher information of the phase of the sinusoidal object per μm² (unit: rad⁻²·μm⁻²). The simulated conditions were set with a field of view (FOV) of 10 μm × 10 μm (16.4 AU × 16.4 AU), an NA of 1.4, a wavelength of 0.7 μm, an immersion media with a refractive index (n) of 1.5, and a camera pixel size of 0.1 μm (0.16 AU).

## Achievable resolving power in the presence of noise

In IbR, photons are the information currency for the resolving power. In extreme cases where no photons are detected, no images—and thus no information—are formed for any imaging modality. In contrast, with unlimited photons where noise is negligible, the IbR of various imaging modalities approach their conventional (Abbe) resolution limits (Supplementary Fig. 5). With a limited photon (defined as the number of photons emitted per area, hereafter referred to as photon density), these modalities fail to achieve their traditional resolution limit due to

Poisson noise. Instead, their practical resolving power differs drastically, and is influenced by the complexity of the imaging target (e.g., spatial frequency of the target) and their photon collection efficiency. By employing our methodology, we evaluated the resolving power of four prevalent imaging modalities—wide-field microscopy, confocal microscopy, two-beam SIM, and ISM—across scenarios involving both planar and volumetric specimens. We calculated information densities for objects with various spatial frequencies ranging from 0 to $\frac{4NA}{\lambda}$.

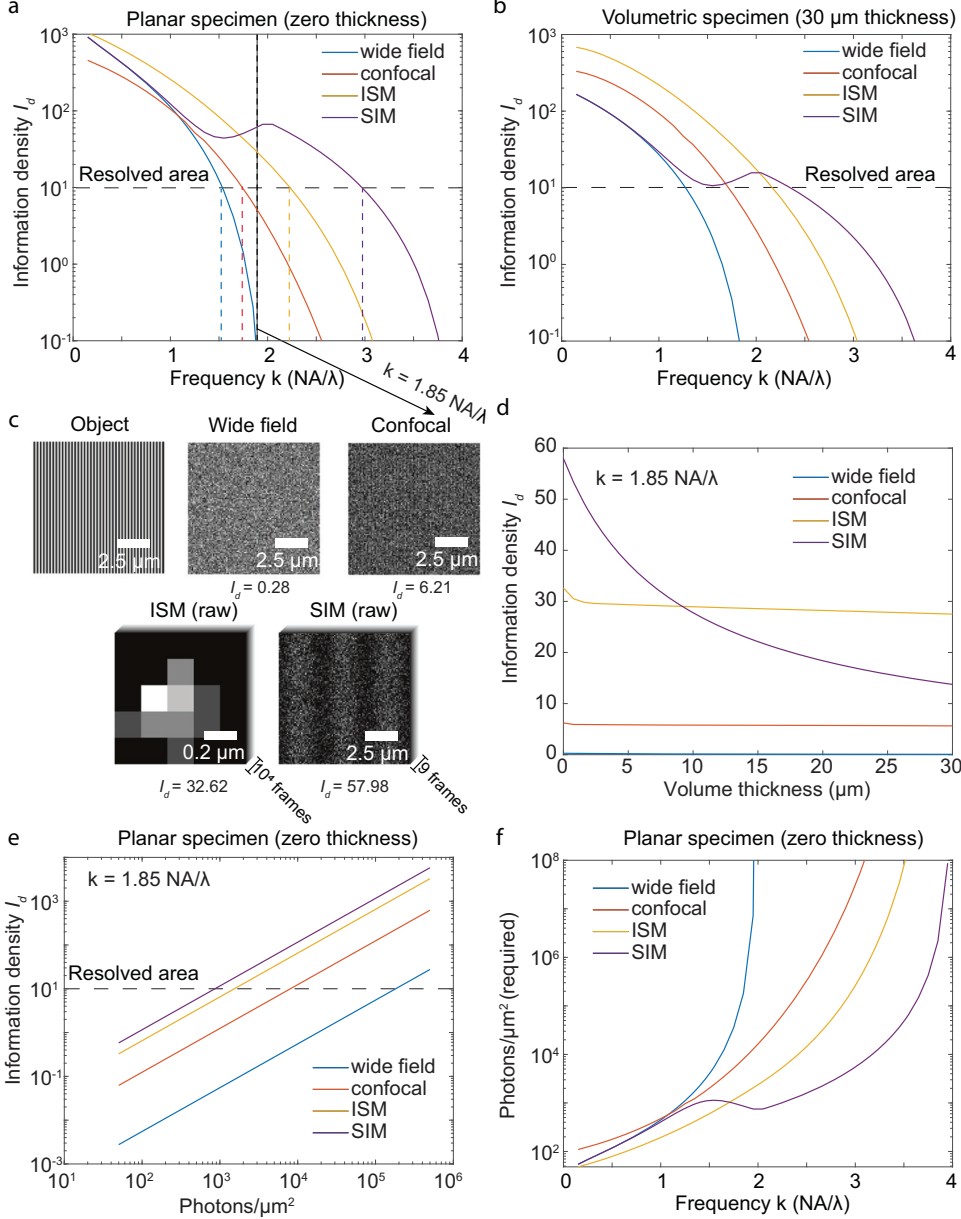

**Fig. 2 | Comparative analysis of resolving power across four imaging modalities (wide-field, confocal, two-beam SIM, and ISM) based on information density $I_d$.** **a** Resolving power, indicated by $I_d$ from four imaging modalities for the case of planar specimens (no background). The resolvability threshold ($I_d = 10\,rad^{-2} \cdot \mu m^{-2}$) marks the applied resolving criterion. Colored dashed lines correspond to the reciprocal of the information-based resolution (IbR) for each modality. **b** Resolving power in terms of information density $I_d$ in the case of volumetric specimens (30 μm thickness). **c** Visualize noisy images of an object at frequency $\frac{1.85NA}{\lambda}$, slightly lower than the diffraction limit. The raw images in the SIM are the images of shifted frequency of the original sinusoidal grating object, as the structured illumination frequency is at optical

transfer function (OTF) boundary, which would not appear in the image. **d** Influence of volume thickness on $I_d$ for each modality at frequency $\frac{1.85NA}{\lambda}$. **e** Relationship between $I_d$ and photon count for an object at frequency $\frac{1.85NA}{\lambda}$ on a logarithmic scale in planar specimen. **f** Emitted photons per area required for different sinusoidal grating structures to meet the resolving criterion ($I_d = 10\,rad^{-2} \cdot \mu m^{-2}$) in planar specimen, ranging from zero to one hundred million photons per μm². Photon collection efficiencies were considered based on 4Pi solid angle emission, objective NA and the pinhole rejection, if present. The above simulation had photon collecting efficiency 32.05% for wide-field and SIM microscopy, 10.77% for confocal microscopy with a 0.5 AU pinhole, and 24.68% for ISM with a 1.3 AU FOV.

In the case of a planar specimen with zero background, at low frequencies, well within the diffraction limit, ISM achieves significantly superior resolving power than all other modalities, whereas the wide-field system and SIM perform similarly. For example, within a low-frequency range ($[0, \frac{NA}{\lambda}]$), ISM's information density $I_d$ is up to double of that for wide-field and SIM systems (Fig. 2a). This can be explained by ISM's effective usage of small area pixel detectors which extends its OTF[36,37], coupled with a multi-pixel based PSF detection that ensures minimum photon loss. In comparison, at these low frequencies, the confocal system's resolving power is significantly lower for planar samples due to photon loss from pinhole rejection. At a frequency of $\frac{0.2NA}{\lambda}$, the confocal system's $I_d$ is fourfold less than that of a wide-field system (Fig. 2a).

At higher frequencies, the wide-field system encounters the diffraction limit at $\frac{2NA}{\lambda}$, resulting in vanishing information (Fig. 2a). In contrast, confocal, SIM, and ISM demonstrate the capacity to exceed the wide-field system's diffraction limit (Fig. 2a). SIM shows the highest resolving power given a limited photon level. With a photon density of 5000 photons/μm² (Fig. 2a), SIM achieves an IbR of 171 nm, exceeding the conventional diffraction-limited resolution of 250 nm. Meanwhile, ISM, confocal, and wide-field systems reach IbRs of 221 nm, 285 nm, and 320 nm, respectively (color dashed line in Fig. 2a). SIM's superior performance is due to its ability to shift the frequency components of the object from outside the OTF boundary to its center, thereby enhancing the magnitude of frequency transfer (Supplementary Fig. 12, 13).

The above results can also be demonstrated from the view of photon emission requirement. To achieve a specific practical resolution (IbR), the minimum numbers of photons required for different imaging modalities drastically defer (Fig. 2f). To resolve an object in a planar specimen with a frequency as low as $\frac{0.2NA}{\lambda}$, a confocal system requires 100 photons/μm², whereas other systems need fewer than 40 photons/μm²—a more than twofold difference. At a frequency of $\frac{NA}{\lambda}$, ISM requires 175 photons/μm², in contrast to other modalities that need at least 350 photons/μm². At a higher frequency of $\frac{1.85NA}{\lambda}$ (slightly below the diffraction limit), the photon demand for a wide-field system soars to 181,336 photons/μm², while a confocal system requires 8045 photons/μm²—about 20 times fewer. ISM and SIM are even more photon-efficient, needing only 1532 photons/μm² and 862 photons/μm², respectively. At significantly higher frequencies, such as $\frac{3.5NA}{\lambda}$, SIM proves to be the most efficient, requiring only 35,783 photons/μm², whereas ISM requires a staggering 32,857,900 photons/μm²—a near three-order magnitude difference. Remarkably, even with one hundred million photons/μm², a confocal system cannot resolve the target structure at this frequency despite its theoretical resolvability based on the optical transfer function of confocal when it has infinite small pinhole[24,28,37].

In the case of volumetric specimens, wide-field and SIM experience significant performance declines due to the increased background from out-of-focus planes in thick samples. Conversely, confocal and ISM, using pinhole for background rejection, maintain information density similar to that of planar specimens. For instance, when imaging an object at a frequency of $\frac{1.85NA}{\lambda}$, as the sample volume thickness increases from 0 to 30 μm, the information density of ISM decreases only slightly from 33 rad⁻²·μm⁻² to 27 rad⁻²·μm⁻². In comparison, the information density of SIM drops drastically from 58 rad⁻²·μm⁻² to 14 rad⁻²·μm⁻², a more than four-fold difference. This superior background resistance of ISM and confocal can be attributed to their optical sectioning capabilities, due to the use of pinholes[9,29,38,39].

In the case of volumetric specimens, both ISM and confocal demonstrate superior resolving power than wide-field and SIM when the frequency of the object is relatively low compared to the diffraction limit. For example, within the low-frequency range ($[0, \frac{NA}{\lambda}]$), ISM

achieves an information density $I_d$ that is up to three times higher than that of the wide-field system and SIM (Fig. 2b), while confocal achieves $I_d$ up to 1.5 times higher than that of the wide-field system and SIM. As the frequency of the target object increases, SIM demonstrates its supreme resolving power at higher frequencies compared to others due to its unique ability to shift the object's frequency component. In the 30 μm volumetric specimen with a signal photon density of 5000 photons/μm² and a background photon density of 500 photons/μm³, the IbR of SIM is 217 nm, whereas ISM, confocal and wide-field systems achieve IbR of 227 nm, 294 nm, and 400 nm, respectively (Fig. 2b).

In terms of photon emission requirement in volumetric specimens, to resolve an object within a volumetric specimen at a low frequency of $\frac{0.2NA}{\lambda}$, ISM and confocal systems require 78 photons/μm² and 155 photons/μm², respectively. In contrast, SIM and wide field need 330 photons/μm², which is four times the requirement for ISM and twice that for the confocal system. At the frequency of $\frac{NA}{\lambda}$, ISM requires 252 photons/μm², and the confocal system requires 602 photons/μm², whereas wide-field and SIM systems require at least 1934 photons/μm². At a higher frequency of $\frac{1.85NA}{\lambda}$ (slightly below the diffraction limit), a wide-field system requires 725,403 photons/μm², and a confocal system needs 8866 photons/μm²—approximately 100 times less. ISM and SIM are more photon-efficient, requiring only 1817 photons/μm² and 3635 photons/μm², respectively. At a significantly higher frequency of $\frac{3.5NA}{\lambda}$, SIM requires the least photon density at 268,512 photons/μm², in contrast to the ISM's requirement of 279,268,000 photons/μm², a difference of more than three orders of magnitude (Supplementary Fig. 6).

For readers who are interested in directly estimating the resolving power given a specific photon count detected, we invite them to examine the surface plot of information density versus frequency and the number of detected photons (Supplementary Fig. 7).

## Effects of numerical aperture (NA) and emission wavelength on the resolving power

While Abbe's resolution limit concludes numerical aperture (NA) and emission wavelength are equally important, their influence on IbR for noise-considered resolution, however, differs. Beyond affecting the OTF, an increase in NA also increases photon collection angle, positioning NA as a more critical factor than wavelength in IbR. For instance, within ISM, when quantifying an object at a frequency of $\frac{1}{0.4}\ \mu m^{-1}$ through the information density $I_d$, an increase in NA from 0.7 to 0.8 results in 2.1 times increase in $I_d$. In contrast, changing the detection wavelength from 0.8 to 0.7 μm yields only 1.5 times increase (Fig. 3a, b).

With a given photon density, IbR can be utilized to calculate the minimum requirement of NA and emission wavelength for different imaging modalities to resolve an object. Considering an instance where a sinusoidal grating object with a frequency of $\frac{1}{0.4}\ \mu m^{-1}$, a photon density of 5000 photons/μm² and an emission wavelength at 700 nm, a wide-field system can theoretically resolve this object with an NA greater than 0.9. Accounting for noise, however, a wide-field system will require at least an NA of 1.19 and a confocal system will require an NA of 1.15, necessitating an oil or water immersion objective. In contrast, ISM and SIM are less demanding, with minimal NAs of 0.85 and 0.67, respectively (Fig. 3a). With a fixed microscope NA of 0.8 (in air), to resolve the structure, the required emission wavelength for wide-field, confocal, ISM, and SIM systems must be smaller or equal to 480 nm, 510 nm, 680 nm, and 890 nm, respectively (Fig. 3b).

## Influence of pinhole size on confocal microscope
In confocal imaging, shrinking pinhole size affects the resolving power in two opposite ways—improving it by broadening effective OTF, while worsening it by decreasing photon detection due to photon rejections of the pinhole[24,37,39,40] (Supplementary Note 4, Supplementary Fig. 11). A too large pinhole diameter, such as 2 AU, yields an extended OTF

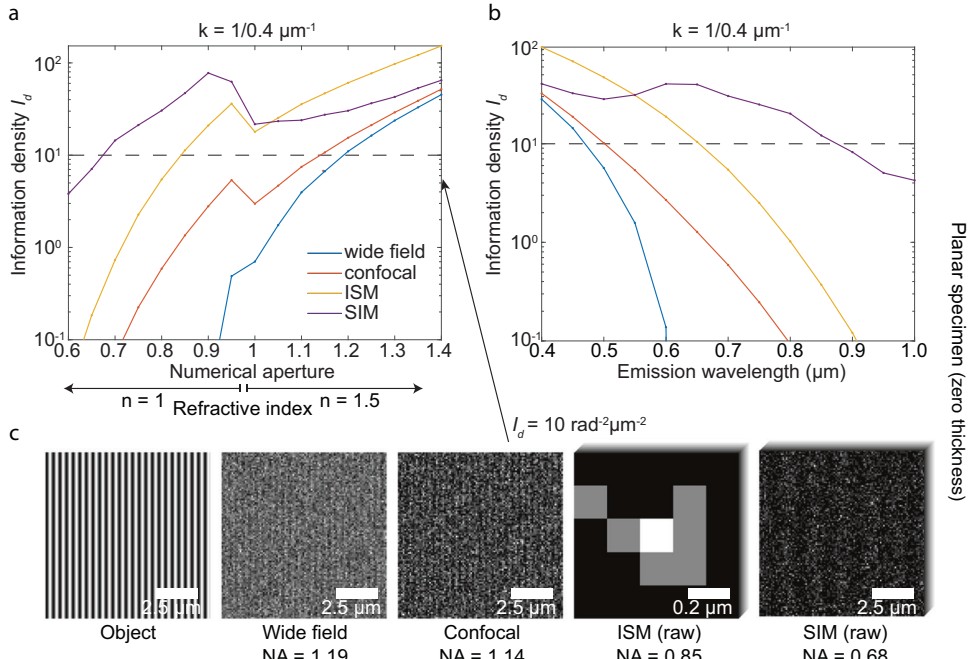

**Fig. 3 | Influence of objective's numerical aperture and emission wavelength on information density $I_d$. a** The relationship between numerical aperture and $I_d$ with an emission wavelength of 700 nm. **b** The relationship between emission wavelength and $I_d$ with a numerical aperture at 0.8 and a refractive index at 1. **c** Visualizations of simulated noisy images along the resolvability threshold $I_d = 10\,rad^{-2}\mu m^{-2}$ across various imaging modalities. In the above plots, photon emission density was set at 5000 photons/$\mu m^2$ with zero background. The confocal system was configured with a pinhole diameter of 0.5 Airy Unit (AU). ISM was set with a detector pixel size of 0.26 AU with 5 by 5 pixels, covering a 1.3 AU square. SIM employed a structured illumination frequency of $k_{st} = \frac{2NA}{\lambda}$, with nine illumination patterns in 3 illumination orientations, one aligning with the sinusoidal pattern object. Each illumination orientation had three phase patterns. The camera pixel size for wide-field and SIM, as well as the scanning intervals for confocal system and ISM, were set to 0.1 μm (0.16 AU). Photon collection efficiencies were considered based on 4Pi solid angle emission, objective NA and the pinhole rejection, if present. The above simulation had a photon collecting efficiency of 32.05% for wide-field and SIM microscopy, 10.77% for confocal microscopy with a 0.5 AU pinhole, and 24.68% for ISM with a 1.3 AU FOV.

akin to that of a wide-field system. (6, 7). Yet, for a 30 μm thick volumetric specimen, a confocal system with a larger pinhole provides a superior IbR compared to wide field, thanks to its ability to reduce out-of-focus background. For instance, for a volumetric specimen of 30 μm with a signal photon density of 5000 photons/$\mu m^2$ and background photon density 500 photons/$\mu m^3$, confocal systems with pinhole diameter of 2 AU ($AU = \frac{1.22\lambda}{NA}$) result in an IbR of 313 nm versus 384 nm for the wide-field system. As the pinhole diameter shrinks, the confocal system acquires better resolving power: decreasing the pinhole diameter to 1 AU and 0.5 AU improves IbR to 300 nm and 285 nm, respectively (Fig. 4a). However, excessively small pinholes, such as 0.1 AU, significantly deteriorate resolution, leading to an IbR of 833 nm due to photon loss.

The tradeoff between improvement and deterioration of the confocal system's resolving power, often balanced by pinhole size from experience, can now be quantified through information density $I_d$. Our simulation suggests an ideal confocal pinhole diameter between 0.5 AU and 1 AU—in agreement with the common practice in confocal systems[12,29]. In the case of 30 μm thick volumetric specimen, for frequency below $\frac{1.3NA}{\lambda}$, a 1 AU confocal pinhole diameters yields higher $I_d$ than a 0.5 AU diameter. Above $\frac{1.3NA}{\lambda}$, a 0.5 AU diameter is more effective (Figure. S7). To seek an optimal resolving power for an object at specific frequencies, Fig. 4b demonstrates the optimal pinhole diameter to achieve the largest $I_d$ given four sinusoidal grating objects of different frequencies. In the case of a 30 μm thick volumetric specimen, for objects of frequency $\frac{0.5NA}{\lambda}$, $\frac{NA}{\lambda}$, $\frac{1.5NA}{\lambda}$, and $\frac{2NA}{\lambda}$, the optimal pinhole diameters are 1.2, 0.9, 0.75, 0.5 AU, respectively. The selection of the optimal confocal pinhole diameter is influenced by the balance between photon collection efficiency and the effective Optical Transfer Function (OTF) enhancement. This balance is not uniform across all frequencies, which leads to varying optimal pinhole sizes depending on the specific spatial frequencies of the sinusoidal grating being imaged (Fig. 4b). Generally, a small pinhole size suits objects of high frequencies, while a large pinhole size suits objects of low frequencies.

Confocal is well acknowledged for its background reduction capability. Another important, often overlooked advantage is its extension of the effective OTF of the imaging system, which enhances resolution beyond that of wide-field system[28,41]. This can be reflected by our simulation in the planar specimen case: at a signal photon density of 5000 photons/$\mu m^2$, confocal systems with pinhole diameters of 1 AU and 0.5 AU achieve an IbR of 308 nm and 303 nm respectively, outperforming the wide-field system's 400 nm (Supplementary Fig. 9). Across a broad frequency range $[0, \frac{1.5NA}{\lambda}]$, a confocal system with a 1 AU pinhole diameter approaches maximum information density. For frequency above $\frac{1.5NA}{\lambda}$, a 0.5 AU pinhole diameter in confocal microscopy is near optimal for information density (Supplementary Fig. 9).

## Influence of detector pixel size and field of view on ISM
The ISM shares a nearly identical optical design with the confocal microscope while delivering markedly superior resolution. The primary differentiator resides in their photon detection method. Confocal microscopy accumulates all photons from each scanning point using a bucket detector. ISM, on the other hand, utilizes an array detector that constructs an image at every scanning point, with each pixel functioning as a pinhole. The array's central pixel is aligned confocally with the scanning point, whereas the centers of surrounding pixels are misaligned[42]. This unique arrangement both extends the effective OTF and minimizes the photon loss usually incurred due to pinhole rejection in confocal setups[28,42].

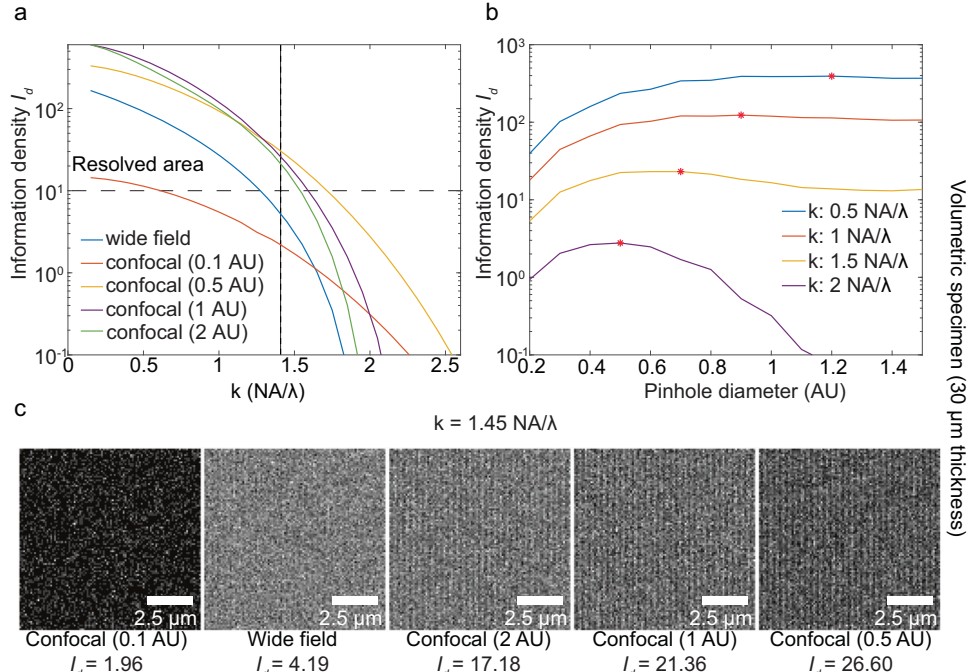

**Fig. 4 | Variation of information density $I_d$ in a confocal system with different circular pinhole sizes. a** Comparisons of information density from a wide-field system and confocal systems with varying pinhole diameters at different target frequencies. **b** The relationship between pinhole diameter and information density $I_d$ for objects at a specific frequency, with the red star denoting the peak $I_d$. **c** Noisy image visualizations for an object frequency of $k = \frac{1.45NA}{\lambda}$. Simulations were performed on a 30 μm thick volumetric specimen, assuming a signal photon emission density of 5000 photons/μm², background photon emission density of 500 photons/μm³, a numerical aperture of 1.4, an immersion medium refractive index of 1.5, and an emission wavelength of 0.7 μm. Camera pixel size in the wide-field system and scanning intervals in the confocal system were set to 0.1 μm (0.16 AU). Photon collection efficiencies were considered based on 4Pi solid angle emission, objective NA and the pinhole rejection, if present. The above simulation had photon collecting efficiency 32.05% for wide-field, 0.66%, 10.77%, 21.12%, 26.79% for confocal microscopy with a 0.1 AU, 0.5 AU, 1 AU, 2 AU pinhole respectively.

The selection of detector pixel size and FOV parameters, traditionally guided by the effective OTF[37] and empirical methods[43], can now be quantitatively assessed using our defined metric of information density-$I_d$ (Fig. 5). In ISM, assuming negligible readout noise from the detector with a constant camera FOV, it has been observed that a smaller pixel size correlates with increased information density. However, reducing the pixel size beyond 0.26 AU does not yield a significant increase in information density. For example, at a frequency of $k = \frac{2.5NA}{\lambda}$, an ISM with a detector pixel size of 0.26 AU has twice the information density of one with a 0.46 AU pixel size, but only a 20% decrease compared to an ISM with a 0.13 AU pixel size.

In addition, with a fixed detector pixel size, we observed that enlarging the FOV enhances information density, although the notable increase plateaued beyond a FOV size of 1.31 AU (Supplementary Fig. 10). In summary, with ignorable readout noise from the detector, within the diffraction-limited frequency range, an ISM detector with a pixel size under 0.26 AU and a FOV size exceeding 1.31 AU are enough to achieve near-optimal performance, achieving information density up to an order of magnitude higher than that of a confocal system with pinhole diameter 0.5 AU.

## Influence of pixel size on resolving power in wide-field microscope

Although the pixel size of the digital image detector is often considered irrelevant to the conventional resolution limit, it has an impact on IbR. In scenarios with negligible sensor noise (e.g., readout noise), reducing pixel size can significantly increase information density-$I_d$. We observed this trend even when pixel size got smaller than that required by the Nyquist sampling theorem[44,45]. In wide-field microscopy, increasing the pixel size from 0.125 μm (0.2 AU) (Nyquist

sampling pixel size) to 0.2 μm (0.33 AU) can reduce the $I_d$ value from 20 rad⁻²·μm⁻² to 12 rad⁻²·μm⁻², roughly two-fold difference. This result underscores the importance of meeting Nyquist sampling pixel size requirement (Fig. 6). In addition, we investigated IbR in situations of applying a pixel size even smaller than that required by Nyquist sampling. We found that further reducing pixel size to 0.04 μm (0.07 AU) enhanced $I_d$ by a quarter compared to the Nyquist sampling pixel size of 0.125 μm (0.2 AU). The presented findings suggest that grouping pixels—akin to using larger pixel sizes in a microscope—compromises the system's effective resolution under photon-limited conditions, given an ideal scenario of zero camera readout noise. While microscope system essentially performs a low pass filter resulting in an diffraction limited image, pixelization (binning pixels) performs another layer of low pass filter on the image. The final image captured by the camera is thus a result of the image being filtered through these two sequential low-pass filters. The low-pass filter effect of pixelization is weaker compared to the OTF of the microscope system (Supplementary Note 3). Reducing the pixel size could improve the frequency transmission rate and increase the information density. Such improvement is obvious when the frequency is close to the diffraction limit boundary, while less pronounced when frequency is close to DC (Supplementary Fig. 15).

## Influence of the structured illumination frequency on SIM

In an aberration-free system, it is intuitive to assume that complex objects—objects containing high-frequency components—will be harder to resolve. However, SIM presents notable deviations from this trend within certain frequency ranges where structures at higher frequencies are better resolved than those at lower frequencies. For example, when employing SIM with an illumination frequency of

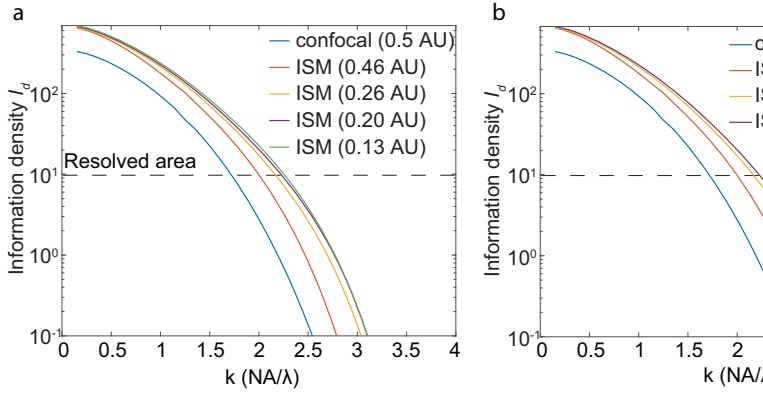

**Fig. 5 | Comparative analysis of detector pixel size and FOV size on resolving power of ISM in terms of information density $I_d$.** The sizes of the detector pixels and the FOV were expressed in terms of their equivalent sizes on the object plane. **a** $I_d$ of a confocal system and ISMs with varying detector pixel sizes, with the FOV size fixed at -1.3 AU. **b** Comparisons of $I_d$ from a confocal system and ISMs with different FOV sizes, with the detector pixel size fixed at 0.26 AU. Other imaging parameters for both setups were the same with a photon emission density of 5000 photons/μm², background photon emission of 500 photons/μm³,

a 30 μm volumetric specimen thickness, a numerical aperture of 1.4, immersion medium refractive index of 1.5, and an emission wavelength of 0.7 μm. Scanning intervals were set to 0.1 μm (0.16 AU) for both systems. Photon collection efficiencies were considered based on 4Pi solid angle emission, objective NA and the pinhole rejection. The above simulation had photon collecting efficiency 10.77% for confocal microscopy with a 0.5 AU pinhole, and 17.81%, 24.68%, 26.04% for ISM with a 0.79 AU, 1.3 AU, 1.83 AU FOV respectively.

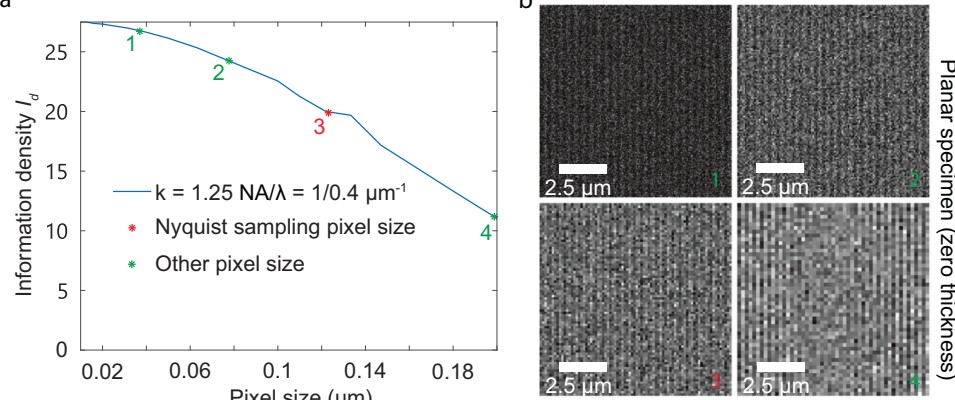

**Fig. 6 | Information density for wide-field microscopy with different camera pixel sizes. a** Impact of camera pixel size on the resolving power in wide-field microscopy in terms of information density $I_d$. Pixel size was denoted as the pixel size mapped on the object plane. Nyquist sampling pixel size is $\frac{\lambda}{4NA}$. **b** Visualization of noisy images at various pixel sizes under a fixed emitted photon count. Conditions of theoretical calculation and simulations included a photon emission density

of 2500 photons/μm², a numerical aperture of 1.4, an immersion medium refractive index of 1.5, and an emission wavelength of 0.7 μm. Photon collection efficiencies were considered based on 4Pi solid angle emission, objective NA and the pinhole rejection, if present. The above simulation had photon collecting efficiency 32.05% for wide-field.

$k_{st} = \frac{2NA}{\lambda}$ (represented by the blue curve in Fig. 7), the information density-$I_d$ at $\frac{2NA}{\lambda}$ is higher, at ~9000 rad⁻²·μm⁻², compared to ~5000 rad⁻²·μm⁻² at $\frac{1.5NA}{\lambda}$. This phenomenon occurs because the emission from the object under structured illumination contains information from both the original and the shifted spatial frequencies. As the object's frequency increases, the original frequency's transfer amplitude diminishes when approaching the OTF boundary. Meanwhile, the shifted frequency's transfer amplitude increases as it nears the OTF center (Supplementary Fig. 12, 13). Since the phase estimation process integrates contributions from both the original and shifted frequencies, the divergent trends of their resulting transfer amplitudes cause the complex relationship between $I_d$ and the object's frequency $k$.

The frequency composition of objects imaged by SIM is influenced by its structured illumination frequency ($k_{st}$). To understand how SIM of different structured illumination would affect its resolving power, we evaluated SIM under four illumination frequencies.

A general trend emerges: higher $k_{st}$ values yield increases $I_d$, and hence better resolving power, across most object frequencies (Fig. 7). This agrees that the commonly used illumination frequency in SIM is at the OTF boundary[25,26]. However, in some cases, a lower illumination frequency can improve the resolving power compared to that of a higher frequency, contradicting this common practice. For example, SIM's information density is lower when using an illumination frequency of $\frac{2NA}{\lambda}$, a frequency at OTF boundary, than that of using a lower illumination frequency (e.g., $k_{st} = \frac{1.4NA}{\lambda}$, Fig. 7a). The above results indicate that in theory using higher structured illumination frequency in SIM is generally beneficial but not always.

In conventional SIM, the structured illumination employs a 1D sinusoidal wave pattern that shifts the object's frequency along a single direction. The common practice is to use a set of three such patterns, oriented at 60-degree intervals, to span the 2D frequency plane. This approach, however, causes an uneven resolution distribution along different directions[25,26]. Quantifying through $I_d$, we measured this

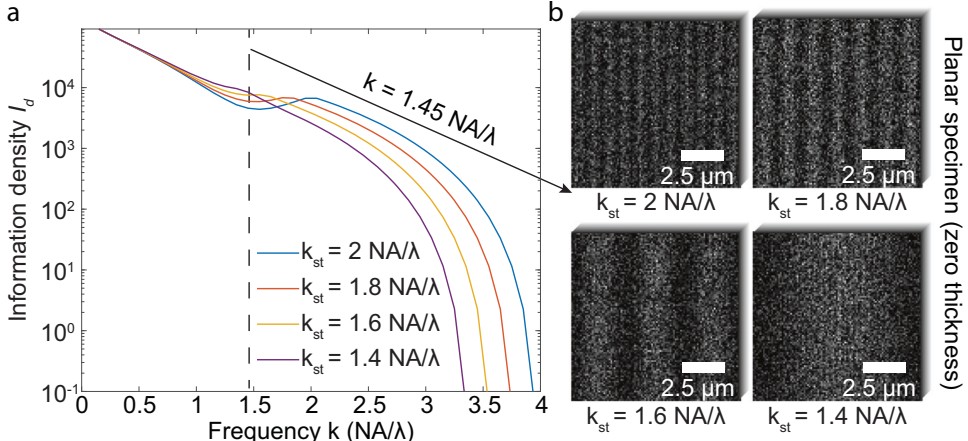

**Fig. 7 | Influence of the structured illumination frequency on SIM performance.**
**a** The information density-$I_d$ of SIM with four structured illumination frequencies were quantified for objects at various frequencies. **b** The visualizations of SIM noisy images were illustrated at different structured illumination frequencies given an object with a frequency of $\frac{1.45NA}{\lambda}$. In the above plots, the photon emission density was set as 5000 photons/μm², with a numerical aperture of 1.4, an immersion medium refractive index of 1.5, and an emission wavelength of 0.7 μm. The SIM

included 9 illumination patterns, divided into 3 illumination orientations with 3 phase patterns each, where one pattern was intentionally aligned with the object's sinusoidal pattern. The camera pixel size was fixed at 0.1 μm (0.16 AU). Photon collection efficiencies were considered based on 4Pi solid angle emission, objective NA, and the pinhole rejection, if present. The above simulation had photon collecting efficiency of 32.05% for SIM microscopy.

orientation-dependent resolution in Supplementary Fig. 13. In the condition of 5000 photons/μm² photon density, we observed that a mere 30-degree alteration in the initial orientation of structured illumination patterns can alter IbR of SIM to vary from 150 nm to 192 nm along a given direction. To alleviate the uneven distribution, we increased the number of illumination pattern orientations. In the above conditions, we tested that utilizing six illumination patterns instead of three resulted in an IbR of 170 nm for all initial orientations (Supplementary Fig. 14).

## Discussion

IbR is designed to establish a noise-considered theoretical resolution limit predicting the performance of imaging modalities with finite photon counts. The concept of IbR relies on the ideal image formation of a periodic object. In our calculation, we assumed the fluorescence response is linear, meaning the emission intensity is proportional to the illumination power. Thus, our IbR is not directly applicable to some of the super-resolution imaging modalities, such as single-molecule localization microscopy (SMLM)[46–49] and stimulated emission depletion (STED) microscopy[50,51], which rely on nonlinear fluorescence response. For example, SMLM requires the stochastic "blinking" of individual emitters. Thus, the resolution limit of SMLM relies on the exact on-off time sequences of imaged single molecules, which is challenging to summarize for IbR. The resolution of STED depends on the power of the depletion laser and its PSF. In an ideal situation where the depletion PSF has a perfect donut shape and infinite power, the resolution of STED can reach the molecule's size[51]. IbR can potentially provide a method for assessing its practical resolution when providing the properties of the non-linear behavior of the probe and its physical model during depletion. In addition, another limitation of IbR is that it only quantifies the lateral resolution in a 2D structure with either planar or volumetric specimens.

While current resolution criteria are mostly defined as the smallest distance at which two closely spaced point objects remain distinguishable. In Abbe's 1873 study, he concluded the resolution expression by examining the visibility of periodic grating structures, not point objects[2]. From a frequency perspective, evaluating resolution with grating structure is more appropriate. A sinusoidal wave structure, for instance, has only one frequency component pair besides the DC component. When these non-DC components surpass

the diffraction limit, the structure vanishes, leading to a complete loss of resolvability. On the other hand, the spatial frequency spectrum of two-point objects extends infinitely. Consequently, even when the distance between two points gets closer beyond the diffraction limit (λ/2NA), there will not be a definitive distance at which they become unresolvable, since the remaining frequency components will still traverse the diffraction barrier (Supplementary Fig. 3).

In modern microscopy, raw data often undergo post-processing to form the image for visualization. It raises the question of whether such post-processing can increase the information and thus the resolution (e.g., IbR). To this end, we provided a theoretical derivation (Supplementary Note 2) showing that deterministic data post-processing methods cannot increase Fisher information. Therefore, post-processing methods, including image reconstruction algorithms, can either keep IbR constant or worsen IbR.

IbR provides a new measure of quantifying the practical resolving power of microscopy imaging modalities considering finite photons. The noise-considered resolution measure offers a theoretical and statistical reference for fluorescence microscope imaging modalities in photon-limited conditions. We believe IbR will become a new concept to provide theoretical guidance for advancement of novel microscopy methods.

## Methods

### 2D sinusoidal pattern generation and microscope imaging system simulation

We simulated a 2D single-tone sinusoidal wave grating object by Eq. (1) in a 20 μm by 20 μm region, which contained 4000 × 4000 pixels, with each pixel representing 5 nm. We generated an ideal image by projecting onto the object plane, achieving unitary magnification. The ideal image formation was processed based on the Fresnel approximation[4], without optical aberrations. This process can be viewed as object convolution with the PSF[4]. To generate the PSF, we started by creating a pupil function of circular shape with radius $\frac{NA}{\lambda}$ in Fourier space. In this space, each pixel denoted a frequency interval of $\frac{1}{L}$, with $L$ equaling the 20 μm length of the simulated image. We then calculated the radius of the pupil function in terms of the number of pixels, which was the rounded value of $\frac{NA}{\lambda} \cdot L - round(\frac{NA}{\lambda} \cdot L)$. The in-focus PSF was the magnitude square of the Fourier transform of this pupil function. Each figure's caption detailed the ideal image's

simulation conditions: numerical aperture of the objective, refractive index of the immersion media, fluorescence emission wavelength, expected photon density, and photon collection efficiency for each imaging modality. In wide-field and SIM microscopes, the photon collection efficiency is determined based on the ratio between solid angle collection and $4\pi$ angle. We assumed the emitter in the sample plane distributes photons uniformly across each direction, represented by a $4\pi$ angle. In confocal and ISM, the photon collection efficiency also accounts for the pinhole effect from physical simulation.

## Wide-field microscopy simulation

In our wide-field microscopy simulation, we obtained the ideal image by directly convolving the object with the PSF and adding the background from the 3D volume. After convolution, to avoid the simulation error at the boundary, we extracted the central 10 μm by 10 μm region from the ideal image to form the FOV. This cropped image was subsequently binned to $100 \times 100$ pixels to mimic the pixel integration effect of camera detection, except for Fig. 6, where the pixel size in the wide-field system was specifically examined. In the $100 \times 100$ pixels image, the camera pixel size was 0.1 μm. The binned image representing $u$ in Eq. (1), was used to further calculate the Fisher information matrix.

## Confocal microscopy simulation

In the confocal system simulation, we assumed a diffraction-limited excitation laser focus, which has the same numerical aperture as the emission detection system. Although the emission wavelength typically exceeds the excitation wavelength due to Stokes shift of the fluorescence emission, we simplified our mathematical calculations by assuming identical wavelengths for both. This assumption allowed us to waive one additional parameter, simplifying the simulation model. As a result, the excitation PSF was identical to the emission PSF throughout our simulation.

On the image plane, we simulated a circular pinhole positioned to conjugate to the center of the excitation PSF. The diameters of the pinhole in each figure were detailed in the figure captions. The image recording sensor was simulated as a bucket photon detector which integrated the collected photons at each scanning position. For each position, we applied a dot product between the object and the excitation PSF to generate the excited object fluorescence distribution. We then convolved this excited object with PSF to form the image before the pinhole. A pinhole mask−with values of 1 inside the pinhole radius and 0 outside−was then dot-multiplied with this image. The sum of this product was assigned to the scanning position. Our simulation encompassed a total of $100 \times 100$ scanning positions to cover the central 10 μm × 10 μm region, with each scanning interval set at 0.1 μm. The image recorded with $100 \times 100$ scanning points representing $u$ in Eq. (1) was used to further calculate the Fisher information matrix.

## Image scanning microscopy simulation

In the ISM simulation, we followed the same steps as that of the confocal system simulation, except for using an array detector instead of a bucket detector with a physical pinhole. We captured images at each scanning position using a camera with a resolution of 5 pixels by 5 pixels, where each pixel measured 0.26 AU. The simulation spanned a total number of 100 by 100 scanning positions, covering a central area of 10 μm × 10 μm (16.4 AU × 16.4 AU). The scanning interval was set at 0.1 μm (0.16 AU). Therefore, in total, 10,000 frames of images of different scanning positions formed the data $u$ in Eq. (1), which was then used to calculate Fisher information.

## Structured illumination microscopy simulation

In SIM, to generate an ideal image, we applied a structured illumination pattern on the object to form a structurally illuminated object. We consistently utilized a single-tone sinusoidal wave pattern. This pattern's intensity oscillated between 0 and 2, thereby ensuring the total emitted photon count remained unchanged. The frequencies of the simulated structured illumination pattern were detailed in the figure captions of the corresponding figures. We convolved the structurally illuminated object with the PSF to form the ideal image. This step paralleled the process in wide-field microscopy, where post-convolution, we selected the central 10 μm by 10 μm portion of the ideal image. This selection was then binned to $100 \times 100$ pixels to mimic the actual camera detection, which had a camera pixel size of 0.1 μm (0.16 AU). In total, we collected images across three structured illumination orientations of 60-degree difference. Each orientation was further divided by three patterns with a phases difference of $\frac{2\pi}{3}$. A total of nine distinct image frames were collected as data $u$ in Eq. (1), which was used for the subsequent Fisher information calculations.

## Background treatment

In the case of planar specimens, we assumed no background, with all fluorescence coming from the target object, which was modeled as infinitely thin within the focus plane. For the volumetric specimen, we simulated a 20 μm by 20 μm by 30 μm volume, comprised of $4000 \times 4000 \times 6000$ pixels (5 nm per pixel). The wide-field approach simulated epi-illumination throughout the entire 3D volume. Similarly, two beam SIM was modeled to create a structured illumination pattern that penetrated the entire 3D volume. Both ISM and confocal were assumed to generate a flawless 3D PSF at each scanning position.

For all imaging modalities, the background that formed on the detector was an overlay of the background simulated from each axial section (section thickness: 5 nm). In our simulation, the number of axial sections was 6000. For each axial section, we calculated the out-of-focus PSF by squaring the magnitude of the Fourier-transformed pupil function after multiplying a defocus factor[52]. Then the background of the axial section was the convolution of the out-of-focus PSF and the uniform distributed fluorophores at the axial section.

## Fisher information calculation

When a random variable (in this case: pixel value) follows Poisson distribution, the Fisher information can be calculated by Eq. (3), which requires the derivative of $u$ with respect to $\phi$. We calculated it numerically by the difference of the $u$ function with a small increment and decrement on the parameter. Specifically, we calculated $\frac{\partial u_i}{\partial \phi} = \frac{u_i(\phi + \Delta\phi) - u_i(\phi - \Delta\phi)}{2\triangle\phi}$, where we set $\Delta\phi$ equal to 0.01 rad in our simulation.

## Data availability

Additional data that support the findings of this study are available from the corresponding author upon request.

## Code availability

Source codes used in numerical calculations are available from the corresponding author upon request.

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

## Acknowledgements

We thank Sheng Liu (University of New Mexico), Cheng Bi, and Hao-Cheng Gao (Purdue University) for their discussions on this topic and Vamara Dembele, Maryam Mahmoodi and Peiyi Zhang from Purdue University for their suggestions during the writing of the manuscript. We also would like to thank Dongyuan Zhou (University of Michigan, Ann Arbor) for scientific proof reading and suggestions. This work was

supported by the US National Institute of General Medical Sciences (R35GM119785).

## Author contributions

F.H. and Y.L. conceived the project. Y.L. developed the mathematical model, carried out the numerical simulations, and wrote the manuscript. F.H. assisted in developing the mathematical model, interpretation of the data, and revised the manuscript.

## Competing interests

The authors declare no competing interests.
