## [Peer Review File · Nature Communications]

A statistical resolution measure of fluorescence microscopy with finite photonsReviewer #1 (Remarks to the Author):

This paper addresses vital questions about resolution in optical microscopes. In particular, it seeks to fill in a gap in our understanding of the practical concept of resolution. The usual resolution theories that are widely used in microscopy are mostly confined to the ideal case, where in effect infinite photons are available to form an image. Realistic advanced microscopies often operate at low photon levels. In this case, the conventional resolution measures are not appropriate. This paper is the first example (at least the first example that this reviewer is aware of) that examines the relative resolution performance of a range of common fluorescence microscopes factoring in the effective photon levels. The methodology is well explained and well justified. The use of Fisher information as the basis of resolution calculations in the presence of noise is highly appropriate. They have confined the analysis to the imaging of single spatial frequency patterns, which seems to me to be the most sensible way to summarise otherwise complex concepts. The comparisons between the four different microscope modalities provides illuminating and non-obvious results, which could have a significant influence on the fundamental understanding of the application of these methods to real imaging challenges. Hence, this paper constitutes an important advance in our understand of microscope resolution and provides a basis for future work, in which the analysis could be expanded to other modalities. I am sure this work will have influence on the field.

There are some suggestions for minor improvements:

An important choice was made, as explained in the sentence "Through visual comparison, we decided to set the threshold of phase estimation uncertainty σ at 0.04 rad as the resolving criterion.". The authors need to explain this choice more carefully, as it is not clear how "visual comparison" was performed. They refer to Fig S2, but there is not enough explanation to understand the basis of their choice.

The written text would benefit from expert proof-reading and editing. While there is no problem with understanding, there are a lot of distracting errors in the text that detract from the overall quality of the content. These are mostly grammatical, such as missing articles or inconsistent tenses.

Reviewer #2 (Remarks to the Author):

This paper introduces a new approach to considering resolution in microscopy, based on Fisher information, which includes the effects of photon noise. So from that point of view it is a significant development. However, we need to ensure that the approach leads to meaningful results, so I think some revisions are necessary.

As the present paper says,

'Inspired by Abbe's experimental approach based on the visibility of the gratings under microscopes ...'
...' Consider a 2D grating object ..'

So this is similar to Abbe's treatment, but Abbe was considering a brightfield system. In fact, fluorescence microscopy was not even invented for another 40 years after his paper. He also included off-axis illumination, so that he describes that interference between two grating orders only is necessary to produce an 'image'. The present paper is confined to symmetrical systems, which seems to encompass most fluorescent microscope systems. Although the need for symmetry is mentioned in the supplementary, a description of how the approach differs from Abbe's earlier on would be useful.
p.4 Line 8. You cite 3 papers on ISM, but not the original paper. Sheppard, Super-resolution in confocal imaging, *Optik* 80, 53-54 (1988).

p.6 Line 18. For clarity mention that α , k , ϕ come from Eq.1.

Fig.2. For SIM the photons are shared between the 9 images? Also, the conditions (pinhole size, SIM frequency, camera pixel size, scanning intervals should be mentioned in the text (p.8) as well as the

caption. Also, there is no mention of the camera size in ISM? I would also like the camera pixel size and scanning intervals to be given in Airy units, to make it easier to see what is going on. How is ISM reconstructed? Usually it's done by weighted averaging in Fourier space. Or maybe direct averaging. Usually SIM also includes a smoothing of the OTF, but I assume you are not doing this, hence the dip in mid-spatial frequencies. The reason that this smoothing is included is that the peak in the OTF for high spatial frequencies results in artefacts in the final image, as a result of a resonance effect at the peak of the OTF. A bit like woofers in hifi give a base that always sounds like the same resonant frequency, regardless of the real frequency. So optimizing for a particular spatial frequency is not the best optimum solution for imaging an object with all spatial frequencies present. Similarly, how is ISM reconstructed? Usually it is done by pixel reassignment (in real space) followed by simple summation. But again it can be done by weighted averaging. Refs 24, 31 are repeated.

In the first part (Fig.2), this is completely a 2D case, which should be mentioned. From the results, a novice would conclude that confocal is not worth doing, whereas there are many confocals around, so there must be some reason. Also, better to state that SIM is 2-beam SIM.

p.9 Background is assumed the same for all systems. What is background? Ref.12 is to Sandison, who describes it as light detected that comes from outside the resolution volume. But you do not treat background in this way. Background is very different in confocal and ISM, as compared with conventional or SIM, because of optical sectioning. I think the results for the effects of background presented are very confusing, because they do not incorporate the effects of optical sectioning. Another paper that describes these effects is that of Gan, Detectability: A new criterion for evaluation of the confocal microscope, Scanning 15, 187-192 (1992), which assumes that background originates from the detected signal from a uniform volume object. This is very different for a confocal system as compared with a conventional system. As you can read from later published papers, background from a volume object is identical for confocal and ISM, if the size of the pinhole and ISM detector are equal. I thought the discussion about the difference between NA and emission wavelength was good, and original. Well done.

The paper goes on to discuss more about particular systems. In my opinion, it would be much more logical to discuss these before comparing different systems, as only then can one appreciate what assumptions are made about different systems.

The conclusion that a pinhole of 0.5 AU is roughly optimum agrees quite well with the results of Sandison and also Gan. This should be mentioned.

The discussions about sampling rate and frequency of SIM illumination frequency are good and original.

p.19 I would like the size of the camera and pixel size for ISM to be also given in Airy units. Especially as camera size is equivalent to pinhole size from the point of view of optical sectioning, as already mentioned.

Supplementary

Fig S6a. The OTF for 1AU has a second lobe. Actually this is negative! So it is very bad for image quality, like in a defocused system. This was described by Gu, Effects of finite-sized detector on the OTF of confocal fluorescent microscopy, Optik 89, 65-69 (1991); Confocal fluorescent microscopy with a finite sized circular detector, J. Opt. Soc. Am. A 9, 151-153 (1992). So 1AU pinhole should really be avoided. 0.5 AU is much better. I am not sure what this would do to Fisher information. Perhaps it would show up in a_{ϕ} , but not in the other measures a_{α} or a_k . It would also be more important if there were more than one spatial frequencies present.

Fig. S6c. This looks very like the curve for integrated intensity for an Airy disk, $1 - J_0^2 - J_1^2$, as given for the collection efficiency for a confocal in many papers. Should mention this?

Fig. S9. For 9 image SIM, there is a substantial variation with grating angle. How do you compare this with other methods in Fig.2?

Fig. S10. For ISM please write FOM in Airy units. 'We found the pixel size of camera in ISM have little influence on ISM's resolving power.' This agrees with previous published works, which should be cited. But it has a big effect on optical sectioning and hence on the effect of background.

Reviewer #3 (Remarks to the Author):

The manuscript by Li and Fang introduces a measure of resolution for microscopy that takes shot noise into account for the resolvable power, opposite to e.g. the Abbe limit that is purely based on the OTF cut off.

The presented idea has a number of problems that partly could be migrated, but overall I think the idea at best will have a small impact into the field. In addition the proposed measure has the unit of angle [rad] which is not intuitive as resolution is given in meters (or 1/meters). Due to the conceptual and detail problems described below I wonder if a more topical journal could be more suited as impact towards users of microscopy seems limited.

Conceptual concerns:

- 1) It is questionable if microscopy is an estimation problem at all and thus the approach to use CRLB is the appropriated formalism. That could be argued, but the problems below remain and are much harder to address.
- 2) The authors suggest to use a sine pattern as resolution target. The found resolution depends on this pattern and very strongly on the periodicity of the pattern.
- 3) The found resolution depends on the FOV, the choice of $10 \times 10 \mu$ FOV is arbitrary.
- 4) The authors suggest a hand-picked resolution threshold of $\sigma_{\phi} = 0.04$ rad. This makes the criterion as good as e.g. the Rayleigh criterion or Sparrow criterion. One might argue that doing an estimation analysis on 2 point emitters with CRLB would be a more natural way to go. This of course has already be done by Ram, Ward, Ober PNAS 2006.
- 5) The found resolution depends on the pixel size in an unexpected fashion. It is beneficial to over sample the image compared to Nyquist sampling. The authors do not discuss this, but it is very strange as the signal is limited to the band limit, but "information" outside the band limit increases the measure. I would expect that sampling at Nyquist gives the wide field resolution for infinite count (but that is sketchy with the hand picked $\sigma_k = 0.04$) and what does then oversampling do?
- 6) The proposed measure has unit of angle. Not spatial frequency or meter, that is not discussed at all, but is neither intuitive nor has been used before.

Detailed issues:

- * Background is not explicitly treated. The background is now hidden in the alpha parameter of the pattern. It appears that σ_{ϕ} is independent of alpha, that is counter intuitive but not discussed.
 - i) is the fisher matrix diagonal? ii) if you simulate a patter as where alpha is very small compared to 1 or larger ($\alpha = 0.01$ or $\alpha = 10$), this changes the count require to see the pattern dramatically. This is not discussed.
- * The authors do not discuss similarity based measures like FRC. Here the noise is taken into account
- * Readnoise is neglected. While this is OK for most of todays scientific cameras it is not even mentioned.
- * The dependence of the resolution of a periodic pattern for estimation was already mentioned in the FRC paper (Nieuwenhuizen 2013 Nat Methods, eq S42). Therefore any picked pattern will be problematic, similarity based measures make this explicit, here this is hidden.
- * Figure 2: The photon count of a confocal is very different than for wide field or SIM at the same photon emission. How is this exactly simulated? Is the sample assumed to be a 0-thickness 2D layer? Does it have 3D structure? How is the background rejected by the pinhole? This matters really a lot a the authors compare modality based on photons, but it is unclear if these are detected photon or emitted photons.
- * In practice a confocal has exactly the same in plane resolution as a wide field system due to the pinhole. At < 1 AU the light efficiency is so bad. Most of the time a biologist would not operate even at 1 AU, but mostly at a larger pinhole to see anything. The author theoretically see this problem (figure S6) but in the main text confocal has a better transfer than wide field for unknown reason.

* Figure 2: ISM should reach the same resolution as SIM for infinite photon count. The OTF of ISM drops off quite a bit faster than in SIM, but that does not matter for infinite SNR. The footprint of the OTF in ISM and SIM is the same.

* The SIM reconstruction influences the structure a lot. In particular the sample itself influences what the deconvolution produces.

* The binning of pixel is of course the same as low pass filtering, please reformulate around l10-15 page 14, this is well known from linear system theory.

* $\Sigma_{\alpha, \sigma}^k$ are not introduced around l19 p6

* the sentence/structure around l20+ p6 is not good, please rewrite.

* Page 3 l 20, the author state the other methods assume band-limited noise? That cannot be true. Noise is structure as read noise, $1/f$ noise, dark noise, shot noise is not band-limited.

* Can the problem be reduced to 1D? if a 2D pattern needs to be resolved, the estimation becomes much harder, more photons, but that is more realistic, as we have a 2D detector not a 1D line detector? This issues is not addressed.

Overall, there are 1) hand-picked threshold for resoleability, 2) hand-picked FOV of $10 \times 10 \mu$ that influences the measure, 3) a 1D periodic structure. This seems too much of subjectivity to ensure a good measure.

**Responses to Reviewers' Comments on
Manuscript: "A statistical resolution measure of fluorescence microscopy with finite
photons"**

We want to thank the reviewers for their time and effort in providing us with comments and suggestions on improving our work. Based on these constructive comments, we have now revised the manuscript with updated simulation models, new data, and figures. Please find a point-by-point response to our reviewers' comments below.

Reviewer #1 (Remarks to the Author):

This paper addresses vital questions about resolution in optical microscopes. In particular, it seeks to fill in a gap in our understanding of the practical concept of resolution. The usual resolution theories that are widely used in microscopy are mostly confined to the ideal case, where in effect infinite photons are available to form an image. Realistic advanced microscopies often operate at low photon levels. In this case, the conventional resolution measures are not appropriate. This paper is the first example (at least the first example that this reviewer is aware of) that examines the relative resolution performance of a range of common fluorescence microscopes factoring in the effective photon levels. The methodology is well explained and well justified. The use of Fisher information as the basis of resolution calculations in the presence of noise is highly appropriate. They have confined the analysis to the imaging of single spatial frequency patterns, which seems to me to be the most sensible way to summarize otherwise complex concepts. The comparisons between the four different microscope modalities provides illuminating and non-obvious results, which could have a significant influence on the fundamental understanding of the application of these methods to real imaging challenges. Hence, this paper constitutes an important advance in our understand of microscope resolution and provides a basis for future work, in which the analysis could be expanded to other modalities. I am sure this work will have influence on the field.

There are some suggestions for minor improvements:

An important choice was made, as explained in the sentence "Through visual comparison, we decided to set the threshold of phase estimation uncertainty σ at 0.04 rad as the resolving criterion.". The authors need to explain this choice more carefully, as it is not clear how "visual comparison" was performed. They refer to Fig S2, but there is not enough explanation to understand the basis of their choice.

Response: We realize this missing rationale in selecting the resolving criterion. In previous work, we selected $\sigma_\phi = 0.04$ as the resolution criterion in an empirical manner, which involves simulations of grating structures from a fixed field of view (FOV) and a subjective decision based on the visualizations. In our updated work, we used Fisher information density I_d as the resolving power measure, eliminating the need for a fixed FOV (**Fig. R1**).

Figure R1. Phase estimation precision limit σ_{ϕ} , Fisher information $I(\phi)$, and Fisher information density I_d in different FOV.

In our updated work, the resolution limit is defined as the reciprocal of the frequency where information density I_d drops below $10 \text{ rad}^{-2} \mu\text{m}^{-2}$ threshold. For an object of $k = \frac{NA}{\lambda}$, applying threshold $I_d = 10 \text{ rad}^{-2} \mu\text{m}^{-2}$ to a unit area (one cycle of the sinusoidal wave), the Fisher information corresponds to phase estimation uncertainty $\sigma_{\phi} = \frac{\pi}{2.2}$. Since $\frac{\pi}{2}$ phase difference can switch sinusoidal pattern to cosine pattern, phase estimation uncertainty of $\sigma_{\phi} = \frac{\pi}{2.2}$ can be a critical point (although still empirical) used as a threshold distinguishing resolved and unresolved. These statements are updated in supplementary notes.

Figure R2. Resolution criterion on information density threshold. The object is at frequency $k = \frac{NA}{\lambda}$. Length of one cycle of the sine pattern is $0.5 \mu\text{m}$, and the pixel size is $0.1 \mu\text{m}$.

We understand that the threshold $I_d = 10 \text{ rad}^{-2} \mu\text{m}^{-2}$ cannot justify all situations and we hope future investigations on the exact critical value could provide a universal resolution criterion for information-based resolution measure.

The written text would benefit from expert proof-reading and editing. While there is no problem with understanding, there are a lot of distracting errors in the text that detract from the overall quality of the content. These are mostly grammatical, such as missing articles or inconsistent tenses.

Response: We thank the reviewer for pointing this out. We have now substantially revised our language and changed our statements for clarity throughout the manuscript.

Reviewer #2(Remarks to the Author):

This paper introduces a new approach to considering resolution in microscopy, based on Fisher information, which includes the effects of photon noise. So from that point of view it is a significant development. However, we need to ensure that the approach leads to meaningful results, so I think some revisions are necessary.

As the present paper says,

'Inspired by Abbe's experimental approach based on the visibility of the gratings under microscopes ...' ...' Consider a 2D grating object ..'

So this is similar to Abbe's treatment, but Abbe was considering a brightfield system. In fact, fluorescence microscopy was not even invented for another 40 years after his paper. He also included off-axis illumination, so that he describes that interference between two grating orders only is necessary to produce an 'image'. The present paper is confined to symmetrical systems, which seems to encompass most fluorescent microscope systems. Although the need for symmetry is mentioned in the supplementary, a description of how the approach differs from Abbe's earlier on would be useful.

Response: We thank the reviewer for pointing this out. We have now added the explanations at the beginning of the results section. The critical point we learned from Abbe is that he accesses the resolution via resolvability on periodic structure gratings instead of two closely spaced points, which most resolution criteria rely on. The reason for using grating to evaluate the resolution rather than two-point emitters can be explained from the frequency point of view. For a sinusoidal wave structure, there is only one pair of frequency components other than DC. When the spatial frequency of the underlying target surpasses the diffraction limit, the contrast of the structure would completely vanish, leading to zero resolvability. However, if using two points, the frequency components spread across the entire spatial frequency spectrum. Therefore, even if the distance of the two points gets closer beyond the diffraction limit ($\lambda/2NA$), it cannot be clearly classified as unresolvable due to its non-vanishing frequency components spreading everywhere within the spatial frequency domain (**Fig. R3**).

Figure R3. Sinusoidal pattern and two close points and their corresponding magnitude of Fourier transform in 1D.

In addition, we agree with the reviewer that brightfield system is significantly different from a fluorescence microscope. We have now added one sentence in the model section specifying the symmetric condition. The brightfield system relies on the transmitted light to form an image, thus, the incident angle would affect the resolution—inclined illumination could allow the system to access frequencies that symmetric illumination cannot. In a fluorescence microscope, the image is formed by the fluorescence, irrelevant to the excitation light angle since the object on the sample plane is now considered as the photon source for image formation.

p.4 Line 8. You cite 3 papers on ISM, but not the original paper. Sheppard, Super-resolution in confocal imaging, Optik 80, 53-54 (1988).

p.6 Line 18. For clarity mention that a , k , f come from Eq.1.

Response: We thank the reviewer's comments. We updated the citation on ISM. After consideration, we have replaced our resolving power measure to Fisher information of phase ϕ per area I_d to further simplify the statement.

Fig.2. For SIM the photons are shared between the 9 images? Also, the conditions (pinhole size, SIM frequency, camera pixel size, scanning intervals should be mentioned in the text (p.8) as well as the caption. Also, there is no mention of the camera size in ISM? I would also like the camera pixel size and scanning intervals to be given in Airy units, to make it easier to see what is going on.

Response: In SIM, the total photons are evenly split into 9 images (3 illumination orientations * 3 phases per illumination pattern). We specified this condition in text and figure captions now. We transferred the detector pixel size and scanning interval of ISM in unit of AU (**Fig. R4**).

Figure R4. ISM FOV and PSF size in unit of AU. For all figures except when ISM FOV and detector pixel size are specially discussed, we use detector pixel size as 0.26 AU and 5 by 5 pixels covering 1.31 AU by 1.31 AU area.

How is SIM reconstructed? Usually it's done by weighted averaging in Fourier space. Or maybe direct averaging. Usually SIM also includes a smoothing of the OTF, but I assume you are not doing this, hence the dip in mid-spatial frequencies. This smoothing is included because the peak in the OTF for high spatial frequencies results in artefacts in the final image, as a result of a resonance effect at the peak of the OTF. A bit like woofers in hifi give a base that always sounds like the same resonant frequency, regardless of the real frequency. So optimizing for a particular spatial frequency is not the best optimum solution for imaging an object with all spatial frequencies present.

Similarly, how is ISM reconstructed? Usually it is done by pixel reassignment (in real space) followed by simple summation. But again it can be done by weighted averaging.

Response: In our work, we *do not* reconstruct images in either SIM or ISM. The Fisher information is directly calculated on the unreconstructed ideal images. The basic unit carrying Fisher information is one pixel. In our simulation, the imaging model for SIM includes 9 ideal structured illuminated images. In ISM, each scanning position generates an ideal image containing 5 by 5 pixels (**Fig. R4**).

We agree with the reviewer that optimizing for a particular spatial frequency does not justify it as an optimum solution for imaging real objects. We added a statement explaining that the optimal resolution determined for a single frequency using IbR should be considered as a reference point rather than a definitive measure. We are still investigating measures to quantify the resolving power for more complex objects.

Figure R5. ISM Fisher information calculation. The Fisher information map is calculated from the image for each scanning position. Final Fisher information is summed together from each information map.

For visual purposes only, **Fig. R6** shows an example reconstruction of an ISM noisy dataset using simple pixel reassignment without filtering^{1,2}. In the supplementary materials, we proved that post-processing, including image reconstruction, would only maintain or decrease the Fisher information content from raw data. Therefore, the reconstructed ISM image may not fully reflect the amount of information contained within a dataset.

Fig. 1. Visualization examples of noise's influence on the resolvability of a single-tone sinusoidal wave grating object. The resolving power of this wide-field system is represented by the information density I_d (unit: $\text{rad}^2\mu\text{m}^{-2}$). Noisy images were simulated with different intensities in this wide-field system at an object frequency of $\frac{1.6\text{NA}}{\lambda}$, with photon densities of $500 \text{ photons}/\mu\text{m}^2$ (left), $5,000 \text{ photons}/\mu\text{m}^2$ (center), and $50,000 \text{ photons}/\mu\text{m}^2$ (right). I_d is the Fisher information of the phase of the sinusoidal object per μm^2 (unit: $\text{rad}^2\mu\text{m}^{-2}$). The simulated conditions were a field of view (FOV) of $10 \mu\text{m} \times 10 \mu\text{m}$ ($16.4 \text{ AU} \times 16.4 \text{ AU}$), an NA of 1.4, a wavelength of $0.7 \mu\text{m}$, an immersion media with a refractive index (n) of 1.5, and a camera pixel size of $0.1 \mu\text{m}$ (0.16 AU).

Refs 24, 31 are repeated.

Response: We thank the reviewer for pointing out this mistake. We removed the duplicate.

In the first part (Fig.2), this is completely a 2D case, which should be mentioned. From the results, a novice would conclude that confocal is not worth doing, whereas there are many confocals around, so there must be some reason. Also, better to state that SIM is 2-beam SIM.

Response: In our updated work, we divide the simulation into two situations: one case for 2D planar specimens and the other case for volumetric specimens where a 2D sinusoidal thin grating is embedded within a $30 \mu\text{m}$ thick specimen with uniform distributed background fluorescence.

Figure R7. Simulation model of volumetric case. A 2D sinusoidal thin grating is embedded in a volume environment with uniform distributed background fluorescence.

With this updated model, we found the performance of wide-field and 2D-SIM deteriorated significantly due to the increase of background from out-of-focus planes in thick specimens, while confocal and ISM resulted in similar information density as compared to the cases of planar specimens.

Figure R8. Information density comparison between imaging modalities in both planar specimen and volumetric specimen. Conditions are identical to Fig.2 in the manuscript.

p.9 Background is assumed the same for all systems. What is background? Ref.12 is to Sandison, who describes it as light detected that comes from outside the resolution volume. But you do not treat background in this way. Background is very different in confocal and ISM, as compared with conventional or SIM, because of optical sectioning. I think the results for the effects of background presented are very confusing, because they do not incorporate the effects of optical sectioning. Another paper that describes these effects is that of Gan, Detectability: A new criterion for evaluation of the confocal microscope, Scanning 15, 187-192 (1992), which assumes that background originates from the detected signal from a uniform volume object. This is very different for a confocal system as compared with a conventional system. As you can read from

later published papers, background from a volume object is identical for confocal and ISM, if the size of the pinhole and ISM detector are equal.

Response: We apologize for the confusion on the background simulation in previous work. In our updated work, we referred to Sandison's and Gan's considerations^{3,4}. In the planar specimen case, we consider there is no background but only the targeting thin 2D sinusoidal grating object. In the volumetric case, we consider the background originated from fluorescence sources uniformly distributed in the 3D volume.

To compare different imaging modalities, we set the total **emitted photon** from the target grating object the same. We agree that the total background fluorescence detected in ISM and confocal would be roughly equal. There could be a minor difference since the ISM detector has a square shape while the confocal pinhole is circular, which is considered in our simulation.

I thought the discussion about the difference between NA and emission wavelength was good, and original. Well done.

Response: We thank the reviewer for the positive comment.

The paper goes on to discuss more about particular systems. In my opinion, it would be much more logical to discuss these before comparing different systems, as only then can one appreciate what assumptions are made about different systems.

Response: We updated the main text structure by introducing more conditions/assumptions about the simulation. In the methods part, we added fully expanded simulation details.

The conclusion that a pinhole of 0.5 AU is roughly optimum agrees quite well with the results of Sandison and also Gan. This should be mentioned.

Response: In previous work, we made a mistake stating a pinhole radius of 0.5 AU as optimal, which, in fact, should be 0.5 AU as the diameter according to our simulations. This is corrected in the latest manuscript. We apologize for such a significant oversight of ours. After correction, we concluded that the optimal confocal pinhole diameter should be between 0.5 AU and 1 AU, depending on the structural complexity. In the volumetric specimen, for frequencies below $\frac{1.3NA}{\lambda}$, confocal with pinhole diameter of 1 AU has greater information density than that with pinhole diameter of 0.5 AU. For frequency above $\frac{1.3NA}{\lambda}$, confocal with pinhole diameter of 0.5 AU tends to have the maximum information (**Fig. R9**).

Fig. S7. Information density I_d in a confocal system with different circular pinhole diameters. (a) Comparison of information density I_d between a wide-field system and confocal systems with different pinhole diameters. (b) Visualization of size comparison between excitation PSF as mapped on the image plane and pinhole. The simulation conditions are in volumetric specimen with thickness of 30 μm set with signal photon emission density of 5000 photons/ μm^2 , background photon emission density 500 photons/ μm^3 , numerical aperture of 1.4, immersion medium refractive index of 1.5, and emission wavelength of 0.7 μm .

The discussions about sampling rate and frequency of SIM illumination frequency are good and original.

Response: We thank the reviewer for the positive comment.

p.19 I would like the size of the camera and pixel size for ISM to be also given in Airy units. Especially as camera size is equivalent to pinhole size from the point of view of optical sectioning, as already mentioned.

Response: We thank the reviewer for the suggestion. We have updated the pixel size in units of AU. We agree with the reviewer on the camera size and pinhole size comment. For ISM, a small detector pixel size favors IbR as it extends the OTF without losing photon count. When the ISM FOV size is equal to the confocal pinhole size, ISM has a larger information density due to its multiple detectors enabling the OTF extension. When the detector pixel size of ISM is equivalent to the pinhole size of the confocal, the central detector pixel in ISM essentially works as a confocal and has a similar information density as confocal⁵. However, ISM would also collect information from peripheral detector pixels. Therefore, ISM always has a greater information content than confocal.

Supplementary

Fig S6a. The OTF for 1AU has a second lobe. Actually this is negative! So it is very bad for

image quality, like in a defocused system. This was described by Gu, Effects of finite-sized detector on the OTF of confocal fluorescent microscopy, Optik 89, 65-69 (1991); Confocal fluorescent microscopy with a finite sized circular detector, J. Opt. Soc. Am. A 9, 151-153 (1992). So 1AU pinhole should really be avoided. 0.5 AU is much better. I am not sure what this would do to Fisher information. Perhaps it would show up in a_{ϕ} , but not in the other measures a_{α} or a_k . It would also be more important if there were more than one spatial frequencies present.

Response: We thank the reviewer for providing this essential information and references. It helped our further investigation on this topic for confocal and ISM theoretically. We agree that the second lobe in OTF would cause serious problems in real object imaging. However, one limitation of our simulation is that we are using a single-tone sinusoidal wave object, which only shows one pair of non-DC components. As a result, the image of this sinusoidal pattern would remain the same even in the presence of the second lobe in OTF.

Through simulation, we noticed that the correlations between a , k , and ϕ are not strong (off diagonal terms in Fisher information matrix are two orders smaller than diagonal term). Therefore, we replaced our resolving power measure with the Fisher information of a single parameter ϕ per area- I_d to simplify the statement.

As we are using single tone sinusoidal wave object, the image is always single tone sinusoidal pattern regardless of the OTF shape. Fisher information is calculated using formula $I(\theta) = \sum_i \frac{1}{u_i} \left(\frac{\partial u_i}{\partial \theta} \right) \left(\frac{\partial u_i}{\partial \theta} \right)^T$, which is directly from the ideal microscopy image. Therefore, the trend for relative amplitude α and ϕ would be the same because varying these parameters would not shift their frequency component positions in the Fourier space. Given the frequency bandwidth of second lobe is narrow and fluctuates, the Fisher information of frequency k could vary depending on the exact position of the Fourier transform of sine pattern. We agree with the reviewer this second lobe should be avoided. It will be an important direction of work, especially when structures with multiple frequencies are used to “probe” the resolvability of a microscopy design. At this stage, we felt the investigation of the Fisher information at second lobe frequency is beyond the scope of this work as its focus is on single-frequency resolvability.

At the same time, we fully agree with the reviewer’s comment. To reflect the resolving power of an imaging system on a generalized specimen would require the evaluation of a more complex object containing multiple spatial frequencies. The specimen structure—the distribution and magnitude/phase of spatial frequencies—affects the resolvability of a particular feature. In this consideration, our current work is limited to evaluating the resolving power of microscopes when a single frequency is present. Expanding IbR towards multi-frequency objects is an exciting and important idea, and we are very interested in this next development step.

Fig. S6c. This looks very like the curve for integrated intensity for an Airy disk, $1 - J_0^2 - J_1^2$, as given for the collection efficiency for a confocal in many papers. Should mention this?

Response: Yes, the photon collection efficiency is using integrated intensity. We simulated it from numerical calculation. We have updated the citations.

Fig. S9. For 9 image SIM, there is a substantial variation with grating angle. How do you compare this with other methods in Fig.2?

Response: In the manuscript’s SIM simulation, we use 9 illumination patterns with 3 illumination orientations, each having 3 phase patterns. One of SIM’s illumination patterns’ orientations aligns with the sine pattern object (0-degree in **Fig. S13**), which gives the optimal information. In the figure caption, we specified the condition that one of SIM structured illumination pattern orientations aligns with the sine pattern structure.

Fig. S10. For ISM please write FOV in Airy units. ‘We found the pixel size of camera in ISM have little influence on ISM’s resolving power.’ This agrees with previous published works, which should be cited. But it has a big effect on optical sectioning and hence on the effect of background.

Response: We have replaced the FOV and detector pixel size in the unit of AU and updated our citations. In our updated work, for a 30 μm volumetric case, we quantified the information density in ISM with respect to detector pixel size and FOV size. We found that given ignorable readout noise from the detector, ISM detector pixel size smaller than 0.26 AU and FOV size greater than 1.31 AU were enough to achieve near optimal performance, yielding up to one order greater information density than confocal in the range of diffraction limit frequency (**Fig. R10**).

Figure R10. Influence of detector pixel size and FOV size on resolving power of ISM in terms of information density I_d . Detector pixel size and FOV size are denoted as the size mapped on the object plane. (a) Information density I_d comparison among a confocal system and ISMs with different detector pixel size when fixing the FOV size around 1.3 AU with respect to target objects’ frequencies. (b) Information density I_d comparison among a confocal system and ISMs with different FOV size when

fixing the detector pixel size 0.26 AU with respect to target objects' frequencies. Conditions are described in Fig. 5.

Reviewer #3(Remarks to the Author):

The manuscript by Li and Fang introduces a measure of resolution for microscopy that takes shot noise into account for the resolvable power, opposite to e.g. the Abbe limit that is purely based on the OTF cut off.

The presented idea has a number of problems that partly could be migrated, but overall I think the idea at best will have a small impact into the field. In addition the proposed measure has the unit of angle [rad] which is not intuitive as resolution is given in meters (or 1/meters). Due to the conceptual and detail problems described below I wonder if a more topical journal could be more suited as impact towards users of microscopy seems limited.

Response: We apologize for the confusion about the resolution unit. The measure σ_ϕ in unit of rad is used to describe the resolving power of the imaging system but not the resolution. Resolution limit was defined as the reciprocal of cut-off frequency at which the resolving power measure σ_ϕ surpass a threshold.

In our updated work, we now use Fisher information of parameter ϕ per area, namely information density I_d to quantify resolving power for a more straightforward understanding. Therefore, we define the resolution limit as the reciprocal of cut-off frequency at which the information density I_d drops below a threshold (**Fig. R11**). This type of resolving criteria are common practices in microscopy field, with example such as Fourier Ring Correlation (FRC) defines resolution as “a cut off frequency at which the cross-correlation value drops below a preset threshold value”⁶.

Figure R11. Information density in wide field microscope system and its optical transfer function. The dashed line indicates the resolution criterion threshold on the information density.

The key difference between our information density and FRC is that information density is a theoretical measure based on the imaging conditions. The noisy images we added to the figures in the manuscript are simulated noisy images for visualization only. They are not used for our Fisher information calculation. Fisher information is purely based on **deterministic** theoretical calculation, and thus, the information density or IbR can be considered as a function of system and sample parameters.

$$I(\phi) = \text{information}(n, NA, \lambda, \text{photon}, k, \text{modality}, d, \text{bg})$$

Where n is the refractive index, NA is the numerical aperture, λ is the emission wavelength, photon is the emitted photons from the target object, k is the frequency of the object, modality represents a particular microscopy design, d is the volume thickness, bg is the background fluorescence density. In contrast, FRC is a post-acquisition resolution evaluation on a specific noisy image.

$$\text{Correlation} = \text{FRC}(\text{acquired_image})$$

The meaning of our theoretical Fisher information measure for resolving power is that it provides a potential way to compare different imaging modalities theoretically in a statistical manner. In Abbe's resolution consideration, resolution is only affected by numerical aperture NA and wavelength λ . Our information density allows us to rewrite resolution as a function of the emitted photons.

Conceptual concerns:

1) It is questionable if microscopy is an estimation problem at all and thus the approach to use CRLB is the appropriated formalism. That could be argued, but the problems below remain and are much harder to address.

Response: We understand the reviewer's concern. Please allow us to explain. As we are trying to incorporate noise into resolution consideration in a theoretical way, we consider Fisher information a rigorous statistical approach to incorporate the randomness of noise into a deterministic measure, which has been widely used to quantify localization precision and sensor noise effect in the field of single-molecule localization microscopy⁷. Fisher information is a concept in statistical theory that measures the amount of information that an observable random variable (pixel values) carries about an unknown parameter (phase of grating). It helps to gauge how changes in the parameter values affect the probability distribution of the data. Essentially, Fisher information quantifies how well experimental data can determine the value of a parameter (phase) in a probabilistic model (Poisson noise). We agree that there might be other ways in examine a noise-considered theoretical resolution measure and we are highly interested in exploring these possibilities as well and potentially compare with the developed IbR.

2) The authors suggest to use a sine pattern as resolution target. The found resolution depends on this pattern and very strongly on the periodicity of the pattern.

Response: We thank the author for bringing up this important consideration. In this work, we consider the periodic sine pattern as a "test probe" for measuring the resolving power of a microscope. In our view, resolution is a property within the imaging system. In this case, we are considering its power in resolving

single tone oscillations with various frequencies similar to that is considered in Abbe’s original paper ‘Contributions to the theory of the microscope and the nature of microscopic vision’⁸, where he observed grating of different frequencies under the microscope.

We quantified the resolvability for which a grating pattern can be distinguished in order to formulate noise-considered resolution measure. However, when noise interferes, it becomes challenging to determine the exact point at which a pattern can be considered resolvable. To address this, our initial approach involved measuring the uncertainty in estimating the phase of a single-tone sinusoidal pattern as a mean to assess resolvability. The rationale behind this method is that if one is unable to accurately determine the phase of a grating (low Fisher information), it implies that the image is not sufficiently clear or resolvable.

Figure R12. Noisy image examples of sine patterns in different intensities and corresponding expectations of information density.

3) *They found resolution depends on the FOV, the choice of 10x10 mu FOV is arbitrary.*

Response: We thank the reviewer for pointing out this in the original submission. In our initial work, we used the estimation uncertainty limit of phase σ_ϕ as the resolving power. Fisher information matrix is additive by its nature. As a result, Fisher information would grow linear with the FOV due to the proportional growth in the number of data points contributing to the Fisher information. Larger FOV result in larger Fisher information and a smaller σ_ϕ . To *eliminate* the requirement of fixed FOV, we replaced σ_ϕ with Fisher information density I_d , **removing the requirement of fixed FOV (Fig. R13)**. Our noise considered resolution is now defined as the reciprocal of cut-off frequency at which the information density I_d drops below a designed threshold.

Figure. R13. Phase estimation precision limit σ_ϕ , Fisher information $I(\phi)$, and Fisher information density I_d in different FOV.

4) The authors suggest a hand-picked resolution threshold of $\sigma_\phi=0.04$ rad. This makes the criterion as good as e.g. the Rayleigh criterion or Sparrow criterion. One might argue that doing an estimation analysis on 2 point emitters with CRLB would be a more natural way to go. This of course has already been done by Ram, Ward, Ober PNAS 2006.

Response: We appreciate the reviewer for providing the information about the paper considering the CRLB of the distance between two points. Our work shares a similarity using Fisher information-related measures in the pursuit to quantify resolution with Ram *et al.*'s work⁹.

From the frequency point of view, we consider using grating to evaluate the resolution is more appropriate in our case. For a sinusoidal wave structure, there is only one pair of frequency components other than the dc component. When structural frequency surpasses the diffraction limit, the microscopy image of the structure would completely vanish leading to zero resolvability as a uniform distribution of photons across the FOV. However, the frequency components of two points would have an infinite span across all spatial frequency domains (**Fig. R14**). Even if the distance of the two points gets closer beyond the diffraction limit ($\lambda/2NA$), there will not be a clear critical point where the information transmitted through the low-pass filter system vanishes.

Figure R14. Figures of single-tone sinusoidal pattern and two-point emitters with the corresponding magnitude of their Fourier transform.

The submitted manuscript uses a single-tone periodic pattern as a “test probe” for measuring the resolving power of a microscope for a particular frequency and thus allowed us to define a clear resolution limit on the different imaging modalities. For example, if imaging a sine pattern of frequency $k = \frac{2.1NA}{\lambda}$, wide field microscopy would not be able to resolve this structure even with infinite photons (**Fig. R18**).

5) They found resolution depends on the pixel size in an unexpected fashion. It is beneficial to oversample the image compared to Nyquist sampling. The authors do not discuss this, but it is very strange as the signal is limited to the band limit, but “information” outside the band limit increases the measure. I would expect that sampling at Nyquist gives the wide field resolution for infinite count (but that is sketchy with the hand-picked $\sigma_k=0.04$) and what does then oversampling do?

Response: We agree with the reviewer that ‘sampling at Nyquist gives the wide field maximum resolution for infinite photon count. When there are infinite photons, wide field microscope would reach its resolution at $\frac{2NA}{\lambda}$, and oversampling would not improve resolution. In the manuscript, we were using an object of a fixed frequency $k = \frac{1}{0.4\mu m}$, where $\frac{2NA}{\lambda} = \frac{1}{0.25\mu m}$. This is a frequency within the diffraction limit.

Although the object’s frequency is within the diffraction limit, due to the influence of noise, it may not be resolved. Quantifying through information density, we discovered that oversampling in this case would increase information density. The images below are simulated with identical total photon counts but with different pixel sizes (**Fig. R15**).

Figure R15. (a) Camera pixel size influence on resolving power of wide-field microscopy in terms of information density I_d . (b) Example visualizations for the same target imaged with different pixel size. Pixel size is denoted as the pixel size mapped on the object plane.

The reason for this information improvement is the low pass frequency filter effect of pixelization, as the reviewer pointed out. Using smaller pixels could allow higher transmission efficiency on non-DC frequencies and this continues when shrinking pixel size beyond Nyquist sampling requirement. Since our target object is a sine pattern, higher frequency transmission efficiency results in increased Fisher information.

6) The proposed measure has unit of angle. Not spatial frequency or meter, that is not discussed at all, but is neither intuitive nor has been used before.

Response: We apologize for the confusion about the resolution unit. The measure σ_ϕ in unit of rad describes the resolving power of the imaging system but not the resolution. The resolution limit was defined as the reciprocal of cut-off frequency which the resolving power measure σ_ϕ surpass a threshold similar to how FRC's defines its image-based resolution.

Detailed issues:

** Background is not explicitly treated. The background is now hidden in the alpha parameter of the pattern. It appears that sigma_phi is independent of alpha, that is counter intuitive but not discussed. i) is the fisher matrix diagonal? ii) if you simulate a patten as where alpha is very small compared to 1 or larger (alpha=0.01 or alpha=10), this changes the count require to see the pattern dramatically. This is not discussed.*

Response: We appreciate the reviewer for pointing out the flaws of our background treatment. Reviewer2 has raised the same concern. In our updated work, we divided the simulation into two situations: one case for 2D planar specimen without background and the other case for volumetric specimens where a 2D

sinusoidal thin grating is embedded within a 30 μm thick specimen with uniformly distributed background fluorescence.

Figure R16. Simulation model in volumetric specimen case. A 2D sinusoidal thin grating is embedded in a volume environment with uniform distributed background fluorescence.

As the reviewer point out, α is the ratio between signal and background. Now we isolated background from signal with independent parameters describing their intensity: signal fluorescence density and background fluorescence density. We stopped using α . To further simplify the model, we reduce to single parameter phase ϕ and use the Fisher information of ϕ per area I_d as the new resolving power measure.

** The authors do not discuss similarity based measures like FRC. Here the noise is taken into account*

Response: We apologize for not discussing the similarities and differences between FRC and our information density. We have updated our introduction and discussion about FRC and clarified their conceptual differences. A response to this question is included in the first portion of our responses to reviewer 3's comments. We appreciate the constructive suggestion from the reviewer.

** Readnoise is neglected. While this is OK for most of todays scientific cameras it is not even mentioned.*

Response: This is a very interesting consideration. Since IbR incorporates photon statistics for resolving power, it is now possible to convert sensor noise level into the resolution metric. A large sensor noise level reduces resolving power, while lower noise cameras improve the achievable resolution at a particular photon-background condition. We felt this investigation can be expanded into multiple directions including comparisons of current sCMOS/CMOS technology with EMCCDs as well as PMT or SPAD arrays of multiple configurations of readout noise and quantum efficiency¹⁰. In this study, we have specified that the readout noise of camera is considered negligible (updated in the text). We are working on the next iteration of our noise-considered resolution where the developed IbR distinguishes camera sensors based on their noise and QE at various photon levels. We thank the reviewer for pointing this out.

** The dependence of the resolution of a periodic pattern for estimation was already mentioned in the FRC paper (Nieuwenhuizen 2013 Nat Methods, eq S42). Therefore any picked pattern will be problematic, similarity-based measures make this explicit, here this is hidden.*

Response: We understand the reviewer's concern. We want to mention that the key differences between our information density measure and FRC is that information density is a theoretical function of imaging system and sample parameters while FRC is an image-based post-acquisition quantification on noisy images.

The Fourier transform of periodic patterns produces discrete frequency components, and FRC operates by analyzing the correlation within passbands in the frequency domain. Consequently, when applying FRC to a periodic pattern, the data points on the correlation versus frequency curve obtained from FRC are predominantly influenced by noise components as opposed to the signal of interest.

In our Fisher information calculation, each periodic sine pattern is used as a unique "test probe" to gauge a microscope's resolving ability at specific frequency and photon density. Every periodic sine pattern contributes a **single** data point to our curve of Fisher information density versus frequency. This curve is constructed by employing multiple sine patterns to probe the imaging system's capabilities.

** Figure 2: The photon count of a confocal is very different than for wide field or SIM at the same photon emission. How is this exactly simulated? Is the sample assumed to be a 0-thickness 2D layer? Does it have 3D structure? How is the background rejected by the pinhole? This matters really a lot as the authors compare modality based on photons, but it is unclear if these are detected photon or emitted photons.*

Response: In our previous work, we assumed the target object was 0-thickness 2D sine pattern grating. In our updated work, we divide the simulation into two situations: one case for 2D planar specimens without background and the other case for volumetric specimens where a 2D sinusoidal thin grating (0 thickness) is embedded within a 30 μm thick specimen with uniform emitted background fluorescence. Specifically, we simulated fluorescence signals from every axial section of the 3D volume on the hypothetical detector. The final background is the summation of all background signals from the entire 3D volume. For the signal photon, when comparing between imaging modalities, we set the emitted photons from the structure (grating) constant. We have specified this in each result section in the revised manuscript.

** In practice a confocal has exactly the same in plane resolution as a wide field system due to the pinhole. At <1 AU the light efficiency is so bad. Most of the time a biologist would not operate even at 1 AU, but mostly at a larger pinhole to see anything. The author theoretically see this problem (figure S6) but in the main text confocal has a better transfer than wide field for unknown reason.*

Response: If the pinhole diameter is greater than 2AU, confocal in plane resolution is the same as a wide field system as their OTF are similar. If the pinhole diameter is smaller than 1AU, the extension of effective OTF in confocal allows higher frequency to pass through where wide field system cannot. Their in-plane resolution differs.

We agree shrinking the pinhole reduces light efficiency drastically. It becomes a tradeoff between improvement brought by extension of OTF and reducing of light collection efficiency. This pinhole size can be quantified through our Fisher information density. In the case of a 30 μm thick volumetric specimen with a signal photon emission density of 5000 photons/ μm^2 and a background photon emission density of 500 photons/ μm^3 , the optimal confocal pinhole diameter in our situations should be between 0.5 AU and 1 AU depending on the structural complexity. In the volumetric specimen, for frequencies below $\frac{1.3NA}{\lambda}$, confocal with pinhole diameter of 1 AU has maximum information density. For frequency above $\frac{1.3NA}{\lambda}$, confocal with pinhole diameter of 0.5 AU tends to have the maximum information density (**Fig. 4**).

Figure 2: ISM should reach the same resolution as SIM for infinite photon count. The OTF of ISM drops off quite a bit faster than in SIM, but that does not matter for infinite SNR. The footprint of the OTF in ISM and SIM is the same.

Response: We agree with the reviewer SIM and ISM would have same resolution at $\frac{\lambda}{4NA}$ when there are infinite photons (**Fig. R17**). However, when photons are limited, their resolution differs due to noise (**Fig. 2a**). Our information density quantifies their resolving power. In the figure below, we simulated the information density curve with photon density of 50 trillion photons per μm^2 to approximate infinite photon case. Wide field would reach resolution $\frac{\lambda}{2NA}$, which converges with the conventional diffraction limit resolution. ISM and SIM approach resolution close to $\frac{\lambda}{4NA}$ given our resolve criterion threshold (dashed line). Our Information based Resolution (IbR) converges with the theoretical resolutions of these imaging modalities when photons are unlimited.

Figure R17. Wide-field, SIM, ISM information density at 50 trillion photons per μm^2 . Wide-field has vanishing information at $\frac{2NA}{\lambda}$. Confocal, ISM, and SIM are approaching $\frac{4NA}{\lambda}$.

** The SIM reconstruction influences the structure a lot. In particular the sample itself influences what the deconvolution produces.*

Response: We understand the reviewer’s comment. Reviewer 2 has a question similar to how we are reconstructing SIM. In our work, we do not reconstruct images in SIM or ISM, thus deconvolution is not involved. The Fisher information is directly calculated on the unreconstructed data. The basic unit carrying Fisher information is one pixel in the idea data frames. In our simulation, the imaging model for SIM includes 9 ideal structured illuminated images. Fisher information is calculated from the summation of 9 Fisher information maps (**Fig. R5** shows the information calculation in ISM).

** The binning of pixel is of course the same as low pass filtering, please reformulate around 110-15 page 14, this is well known from linear system theory.*

Response: We appreciate the reviewer's feedback. In response, we have revised our wording. While the concept of low-pass filtering in the context of pixel binning is widely recognized, this understanding is crucial in explaining why oversampling can enhance Fisher information. This is due to the smaller pixel sizes allowing a greater passage of higher frequencies even beyond Nyquist sampling requirement.

** Sigma_alpha, sigma_k is not introduced around 119 p6*

Response: We have revised our simulation, reducing the estimation parameters to phase ϕ . Now we are using Fisher information of ϕ per area, namely information density I_d to quantify resolving power.

** the sentence/structure around l20+ p6 is not good, please rewrite.*

Response: We reformulated the sentences with updated simulation model.

** Page 3 l 20, the author states the other methods assume band-limited noise? That cannot be true. Noise is structure as read noise, 1/f noise, dark noise, shot noise is not band-limited.*

Response: We apologize for not being specific and revised this part. The method we refer to that assumes band-limited noise is the information capacity approach by Cox *et al*¹¹. Their approach quantifies the resolution of imaging systems from a point assuming the number of freedoms is constant. $N_F = 2(1 + L_x B_x)(1 + L_y B_y)(1 + T B_T)(\log(1 + \frac{S}{n}))$ where B_x, B_y are the spatial bandwidths, L_x, L_y are the dimensions of FOV in x,y directions, T is the observation time, B_T is the temporal bandwidth of the optical system, and $\frac{S}{n}$ is the signal-to-noise ratio where noise is assumed to be bandlimited as spatial and temporal as B_x, B_y, B_T .

** Can the problem be reduced to 1D? If a 2D pattern needs to be resolved, the estimation becomes much harder, more photons, but that is more realistic, as we have a 2D detector not a 1D line detector? This issue is not addressed.*

Response: This problem can be reduced to 1D. We developed this work in 1D first and then extended it to 2D structure. One important difference between 1D and 2D simulation is on the OTF. In 1D, the OTF has a triangle shape from auto-correlation of square pupil function. In 2D, OTF is the auto-correlation of the circular pupil function. The center cross-section line profile of 2D OTF differs from 1D (**Fig. R18**). In our updated work, we also included the 3D volumetric background noise. We maintained our focus on 2D resolution evaluations.

Figure R18. The OTF of 1D case and center cross-section line profile of a 2D OTF.

Reference

1. Müller, C. B. & Enderlein, J. Image Scanning Microscopy. *Phys Rev Lett* **104**, 198101 (2010).
2. Sheppard, C. J. R., Mehta, S. B. & Heintzmann, R. Superresolution by image scanning microscopy using pixel reassignment. *Opt Lett* **38**, 2889–2892 (2013).
3. Gan, X. S. & Sheppard, C. J. R. Detectability: a new criterion for evaluation of the confocal microscope. *Scanning* **15**, 187–192 (1993).
4. Sandison, D. R. & Webb, W. W. Background rejection and signal-to-noise optimization in confocal and alternative fluorescence microscopes. *Appl Opt* **33**, 603–615 (1994).
5. Sheppard, C. J. R. *et al.* Pixel reassignment in image scanning microscopy: a re-evaluation. *JOSA A* **37**, 154–162 (2020).
6. Koho, S. *et al.* Fourier ring correlation simplifies image restoration in fluorescence microscopy. *Nat Com* **10**, 3103 (2019).
7. Chao, J., Ward, E. S. & Ober, R. J. Fisher information theory for parameter estimation in single molecule microscopy: tutorial. *JOSA A* **33**, B36–B57 (2016).
8. Abbe, E. Beiträge zur Theorie des Mikroskops und der mikroskopischen Wahrnehmung. *Archiv für Mikroskopische Anatomie* **9**, 413–468 (1873).
9. Ram, S., Ward, E. S. & Ober, R. J. Beyond Rayleigh's criterion: a resolution measure with application to single-molecule microscopy. *Proceedings of the National Academy of Sciences* **103**, 4457–4462 (2006).
10. Ober, R. J., Ram, S. & Ward, E. S. Localization accuracy in single-molecule microscopy. *Biophys J* **86**, 1185–1200 (2004).
11. Cox, I. J. & Sheppard, C. J. R. Information capacity and resolution in an optical system. *JOSA A* **3**, 1152–1158 (1986).
12. Kintner, E. C. An analytic recurrence procedure for computing the cross-multiplication coefficients in an analytic OTF method. *Optica Acta: International Journal of Optics* **24**, 1237–1246 (1977).
13. Kintner, E. C. & Sillitto, R. M. A new 'analytic' method for computing the optical transfer function. *Optica Acta: International Journal of Optics* **23**, 607–619 (1976).

Reviewer #1 (Remarks to the Author):

I have read the authors' revision comments and the new version of the manuscript. I have considered the responses to my own comments and those to the comments of the other reviewers. I am convinced that this is a valuable piece of work that could provide the basis for many future studies on comparisons of resolutions across microscopes in realistic imaging situations. I am not aware of another method that can perform this useful task.

Reviewer #2 (Remarks to the Author):

I am happy with this paper now, except for two points.

1. 'In previous work, we made a mistake stating a pinhole radius of 0.5 AU as optimal, which, in fact, should be 0.5 AU as the diameter according to our simulations.'

AU is a ratio measurement, i.e. 0.5 AU pinhole radius is 0.5 times the radius of the first zero of the Airy disk. The pinhole diameter is also 0.5 times the diameter of the first zero of the Airy disk. So I am confused by your statement.

2. 'However, one limitation of our simulation is that we are using a single-tone sinusoidal wave object, which only shows one pair of non-DC components. As a result, the image of this sinusoidal pattern would remain the same even in the presence of the second lobe in OTF.'

Even with a single frequency object, if the frequency coincides with the second lobe, surely the sign will be reversed relative to the DC term, i.e. the contrast of the bars is reversed. Not a very accurate image!

Other than these points, I am happy with the manuscript.

Reviewer #3 (Remarks to the Author):

The authors have done a very large effort to revise the paper. They have changed essential parts of the original paper. I am mostly happy with the new approach as the authors have seriously taken the comments of the different reviewers into account.

Still there remain a number of points, some minor, but some also major.

* dependency on the choice of pattern

The authors write they use it "to that is considered in Abbe's original paper". Of course Abbe does not consider noise or SNR at all, therefore a grating is a good choice to understand spatial frequency and it does not matter how spatially extend the pattern is. For an estimation process that is totally different as the SNR determines everything.

* In the limit the measure should reproduce e.g. the Abbe limit and twice that for SIM and ISM

In figure R11 and r17 that is not the case. However, in the text the authors state that everything is as expected. Is the plot wrong? In particular R17, the tick at $4/NA \lambda$ is not reached by quite a bit by both ISM and confocal for infinite count.

* The measure for SIM

The authors state that they do not reconstruct the image. How can they get a higher resolution by summing the Fisher information maps then?? Each image alone as $1/9$ the information as all, but that should be then equal to $9*$ a wide field exposure. I really cannot see that 9 raw SIM frames have a higher resolution than 9 wide field frames because they are equal.

* Threshold value of 10 on information density

In figure R5 there seem to be two different values?

* "In Abbe's resolution consideration, resolution is only affected by numerical aperture NA and wavelength λ . Our information density allows us to rewrite resolution as a function of the emitted photons."

This must be detected photons. This is an essential point, the number of emitted photons does not contribute to the picture in classical image formation. For quantum effects this matters. This also pertains to problems later when comparing e.g. confocal imaging.

* My initial concern "The authors suggest to use a sine pattern as resolution target. The found resolution depends on this pattern and very strongly on the periodicity of the pattern." Was not resolved by the answer. Only if the authors can show that a sine wavelet (a sine not running over the full FOV gives the same resolution as a full sine then I am convinced that this gives a meaningful number).

Later the authors come back to this point that each frequency only contributes a single data point. I do not get the point as an extended pattern gives better SNR in frequency domain than a smaller pattern.

* resolution in a confocal

The signal is given by an emission density e.g. 5000 photon/ μ^2 . But that does not say anything as it is about the detected photons. You need at least a flux to convert the emission density to counts on a detector.

* A very confusing point is "the optimal confocal pinhole diameter in our situations should be between 0.5 AU and 1 AU depending on the structural complexity" How can the value depend on the specimen?? The authors early stated that the resolution does not depend on the sample?

* Oversampling. I think here the authors make a real mistake in their reasoning "... this understanding is crucial in explaining why oversampling can enhance Fisher information. This is due to the smaller pixel sizes allowing a greater passage of higher frequencies even beyond Nyquist sampling requirement."

There are no frequencies above the Nyquist sampling rate at the detector expect noise. At Nyquist the low pass filtering effect of the discretization/pixels is just so that it does not influence the reconstruction of the signal anymore because the signal is band limited by the optics to half the Nyquist rate. That is the whole point of Nyquist frequency of course.

Oversampling does not transfer more information. The system is hard band limited by the acceptance angle of the lens. If the measure produces a better value with oversampling, the authors must make a very, very strong case how this can happen.

Responds to Reviewers' Comments on

Manuscript: "A statistical resolution measure of fluorescence microscopy with finite photons"

We want to thank the reviewers for their time and effort in providing us with comments and suggestions on improving our work. Based on these constructive comments, we have now revised the manuscript with additional content and figures. Please find a point-by-point response to our reviewers' comments below.

Reviewer #1 (Remarks to the Author):

I have read the authors' revision comments and the new version of the manuscript. I have considered the responses to my own comments and those to the comments of the other reviewers. I am convinced that this is a valuable piece of work that could provide the basis for many future studies on comparisons of resolutions across microscopes in realistic imaging situations. I am not aware of another method that can perform this useful task.

Response: We thank the reviewer for the positive comment.

Reviewer #2 (Remarks to the Author):

I am happy with this paper now, except for two points.

1. 'In previous work, we made a mistake stating a pinhole radius of 0.5 AU as optimal, which, in fact, should be 0.5 AU as the diameter according to our simulations.'

AU is a ratio measurement, i.e. 0.5 AU pinhole radius is 0.5 times the radius of the first zero of the Airy disk. The pinhole diameter is also 0.5 times the diameter of the first zero of the Airy disk. So I am confused by your statement.

Response: We appreciate the reviewer's clarification on the Airy Unit. Following this convention, we have now checked the manuscript to ensure its reflection of the consideration of AU as a ratio measurement.

2. 'However, one limitation of our simulation is that we are using a single-tone sinusoidal wave object, which only shows one pair of non-DC components. As a result, the image of this sinusoidal pattern would remain the same even in the presence of the second lobe in OTF.'

Even with a single frequency object, if the frequency coincides with the second lobe, surely the sign will be reversed relative to the DC term, i.e. the contrast of the bars is reversed. Not a very accurate image!

Response: We appreciate the reviewer's clarification on the confocal image formation. The contrast of the sinusoidal pattern image would be reversed with respect to the sinusoidal pattern object when its spatial frequency locates within the second lobe (below we simulate a confocal system with 1 AU pinhole).

Figure R1. Sinusoidal gratings and its contrast in confocal system. (a, b) Confocal image frequency amplitude versus effective confocal OTF in 1D. Dashed black line indicates frequency $k = \frac{2NA}{\lambda}$. (c, d) Comparison between the objects and their confocal images. (e, f) Line profile of objects and their confocal images. The confocal system has a pinhole size of 1 AU.

While the contrast of the image is reversed in confocal when its frequency coincides with the second lobe, we can still calculate the Fisher information by its formulation $I(\phi) = \sum_i \frac{1}{u_i} \left(\frac{\partial u_i}{\partial \phi} \right)^2$.

As the transmission is low on the second lobe ($\sim 0.5\%$), we found the information density is very low and thus makes it difficult to reach the resolvable criterion (Fig. R2).

Figure R2. Information density in confocal with 1AU pinhole. Signal photon emission density is 5000 photons/ μm^2 . The microscope system has a numerical aperture of 1.4, an immersion medium refractive index of 1.5, and an emission wavelength 0.7 μm .

Other than these points, I am happy with the manuscript.

Response: We extend our sincere gratitude to the reviewer for the revision comments and the valuable supporting information provided.

Reviewer #3 (Remarks to the Author):

The authors have done a very large effort to revise the paper. They have changed essential parts of the original paper. I am mostly happy with the new approach as the authors have seriously taken the comments of the different reviewers into account.

Response: We thank the reviewer for the encouraging and constructive review.

Still there remain a number of points, some minor, but some also major.

** dependency on the choice of pattern*

The authors write they use it “to that is considered in Abbe’s original paper”. Of course Abbe does not consider noise or SNR at all, therefore a grating is a good choice to understand spatial frequency and it does not matter how spatially extend the pattern is. For an estimation process that is totally different as the SNR determines everything.

Response: The information density measure is based on imaging sinusoidal patterns as testing object. It certainly depends on the choice of pattern. The Fisher information of phase is calculated in the imaging space instead of frequency space and grows linearly with the number of cycles of sinusoidal pattern contained. When the FOV is large such that the effect of the boundary not containing whole cycles is minimal, using the information of phase per area would minimize the effect related to how spatially extended the pattern is. Here, we demonstrate the intermediate steps of the Fisher information map calculation for the wide-field microscope in **Fig R3**.

Figure R3. Fisher information calculation in wide-field microscope. (a) Ideal image and its derivative plotted in one cycle. (b) Ideal image of a sinusoidal grating object in wide field microscope. (c) Fisher information (unit: rad^{-2}) distribution in one cycle of the sinusoidal pattern. (d) Fisher information map from a wide-field microscope.

Information is calculated by formula $I(\phi) = \sum_x \frac{1}{u(x)} \left(\frac{\partial u(x)}{\partial \phi} \right)^2$. The sinusoidal grating object has 500×500 pixels with pixel size $0.02 \mu\text{m}$, a frequency of $k = \frac{NA}{\lambda}$. Signal photon emission density is $5000 \text{ photons}/\mu\text{m}^2$. The microscope system has a numerical aperture of 1.4, an immersion medium refractive index of 1.5, and an emission wavelength $0.7 \mu\text{m}$. The photon collection efficiency is 32.05%, calculated from solid angle collection.

1,2

** In the limit the measure should reproduce e.g. the Abbe limit and twice that for SIM and ISM*

In figure R11 and r17 that is not the case. However, in the text the authors state that everything is as expected. Is the plot wrong? In particular R17, the tick at $4/NA \lambda$ is not reached by quite a bit by both ISM and confocal for infinite count.

Response: Confocal would only reach twice the resolution of diffraction limit resolution when it has an infinite small pinhole^{1,2} and infinite photon count. When the pinhole is very small, the photon collection efficiency is close to zero such that even a reasonably large number of emitted photons would not generate enough information to be considered resolvable.

In a hypothetical situation, we set the detected photons to be five quadrillion ($5 * 10^{15}$) per μm^2 given a confocal with small pinhole (0.10 AU) and an ISM with small detector pixel size (approached as 0.13 AU). Our results show they are both approaching the $\frac{4NA}{\lambda}$ as resolvable frequency.

Figure R4. Information density in confocal and ISM with large number of photons detected $5 * 10^{15}$ per μm^2 . (a) Information density in confocal of 0.1 AU pinhole. (b) Information density in ISM of detector pixel size 0.13 AU and FOV 1.31 AU.

** The measure for SIM*

The authors state that they do not reconstruct the image. How can they get a higher resolution by summing the Fisher information maps then?? Each image alone as 1/9 the information as all, but that should be then equal to 9 a wide field exposure. I really cannot see that 9 raw SIM frames have a higher resolution than 9 wide field frames because they are equal.*

Response: Each frame of the SIM image is an image of structured illuminated object. The structured illumination, by interacting with the sinusoidal grating's features, enriches the raw dataset with additional spatial information that is not present in wide-field imaging (Fig. R5). As a result, variation of the sinusoidal grating's phase generates changes in the image of structured illuminated object even when the grating's spatial frequency is beyond diffraction limit (Fig. R5).

On the contrary, the ideal image in wide field is uniform (no changes of the image caused by object's phase variation) when the grating's frequency is beyond diffraction limit.

Fisher information is directly calculated from such variation $I(\phi) = \sum_i \frac{1}{u_i} \left(\frac{\partial u_i}{\partial \phi} \right)^2$ where u is the nine ideal images of structured illuminated object in SIM. Below figures demonstrate the ideal image generation and information map in wide-field and SIM when imaging sinusoidal grating objects with different spatial frequencies.

Figure R5. Fisher information calculation in wide-field and SIM. Wide-field and SIM are simulated under the same number of total detected photons. Each frame of SIM image receives 1/9 photons of wide-field image. At frequency $k = \frac{2.4NA}{\lambda}$, wide field has no information because the structure is beyond OTF boundary, SIM's Fisher information results from shifted frequency of the structure falling inside microscope's OTF.

Figure R6. Fisher information calculation in SIM. The Fisher information of SIM is the sum of nine information maps generated from nine images of structured illuminated object. Angles and phases indicate the difference between object and structured illumination pattern.

** Threshold value of 10 on information density*

In figure R5 there seem to be two different values?

Response: Thanks for pointing out this error in Figure R5 in the previous response letter. The line is placed at an incorrect position during our figure generation. We have double checked all figures in the manuscript and confirmed all figures are free of typographical errors.

** “In Abbe’s resolution consideration, resolution is only affected by numerical aperture NA and wavelength λ . Our information density allows us to rewrite resolution as a function of the emitted photons.”*

This must be detected photons. This is an essential point, the number of emitted photons does not contribute to the picture in classical image formation. For quantum effects this matters. This also pertains to problems later when comparing e.g. confocal imaging.

Response: We agree with the reviewer that ‘*the number of emitted photons does not contribute to the picture in classical image formation.*’ The reason we used the emitted photons is to account for the factors that influence the photon collection efficiency, such as numerical aperture and pinhole size in confocal. For instance, confocal with infinite small pinhole would result in largest information density compared to other confocal microscopes if detecting the same

number of photons. However, this is rarely used because the pinhole rejects nearly all emitted photons. Using emitted photon in IbR takes this into consideration.

In the updated work, we added the photon collection efficiency for each imaging modality in the figure caption. We also added the figures of information density versus the detected photons for each imaging modality separately (Fig. R7).

Figure R7. Fisher information calculation in four imaging modalities with respect to detected photons and frequency. Red planes indicated the resolving criterion $I_d = 10 \text{ rad}^{-2} \cdot \mu\text{m}^{-2}$. The simulation conditions were conducted in planar specimen (background free), numerical aperture of 1.4, immersion medium refractive index of 1.5, and emission wavelength of $0.7 \mu\text{m}$. The confocal system was configured with a pinhole diameter of 0.5 Airy Unit (AU). ISM modality was set with a detector pixel size of 0.26 AU with 5 by 5 pixels, covering a 1.3 AU square. SIM employed a structured illumination frequency of $k_{st} = \frac{2NA}{\lambda}$, with nine illumination patterns in 3 illumination orientations, one aligning with the sinusoidal pattern object. Each illumination orientation had three phase patterns. The camera pixel size for wide-field system and SIM, as well as the scanning intervals for confocal and ISM, were set to $0.1 \mu\text{m}$ (0.16 AU).

** My initial concern “The authors suggest to use a sinusoidal pattern as resolution target. The found resolution depends on this pattern and very strongly on the periodicity of the pattern.” Was not resolved by the answer. Only if the authors can show that a*

sinusoidal wavelet (a sinusoidal not running over the full FOV gives the same resolution as a full sinusoidal then I am convinced that this gives a meaningful number).

Response: We chose a large FOV that contains multiple cycles (thirty cycles when $k = \frac{NA}{\lambda}$) such that the information calculation error introduced by not containing integer number of cycles is small comparatively and can be omitted (A more detailed discussion is included above and Figure R3). As an example, the sum of the information in one cycle in Fig. R3 is 1.275 rad^2 for an area of $0.02\mu\text{m} * 0.5 \mu\text{m}$, resulting in an information density of $127.5 \text{ rad}^2\mu\text{m}^{-2}$. The sum of information of the entire FOV is $1.275\text{e}4 \text{ rad}^2$ for the area of $10\mu\text{m} * 10\mu\text{m}$, resulting in information density of $127.5 \text{ rad}^2\mu\text{m}^{-2}$. The information densities are equal.

Later the authors come back to this point that each frequency only contributes a single data point. I do not get the point as an extended pattern gives better SNR in frequency domain than a smaller pattern.

Response: We evaluate the information density at each spatial frequency using a sinusoidal grating object at that specific frequency. These objects contain multiple cycles of the sinusoidal pattern depending on the spatial frequency and FOV of simulation (Fig. R8).

We concur with the assertion that ‘an extended pattern gives better SNR in frequency domain than a smaller pattern’. However, in our methodology, information is computed in the image domain rather than the frequency domain. Excluding the boundary error, which results from not encompassing a complete cycle of the sinusoidal pattern, the information per area remains consistent regardless of the extension of the pattern.

Figure R8. Information density in wide-field microscope. Each condition of spatial frequency on the right generates one data point on the left curve.

* resolution in a confocal

The signal is given by an emission density e.g. 5000 photon/ μm^2 . But that does not say anything as it is about the detected photons. You need at least a flux to convert the emission density to counts on a detector.

Response: In this work, we assumed the emitter is emitting photons in a 4π solid angle, while NA and refractive index n characterize the solid angle from which photons are collected. In confocal and ISM, we also consider the pinhole photon collection efficiency from physical simulation. We have updated this photon flux convert condition in the figure caption and methods description.

Figure R9. Solid angle photon collection efficiency diagram. Angle θ_0 is calculated from $NA = n \sin(\theta_0)$. Photon collection efficiency is calculated by $efficiency = \frac{S}{4\pi r^2} = \frac{\int_0^{\theta_0} \int_0^{2\pi} r^2 \sin(\theta) d\phi d\theta}{4\pi r^2} = \frac{1 - \cos(\theta_0)}{2}$.

Figure R10. Confocal pinhole photon collection efficiency. (a) Confocal microscope diagram. (b) Pinhole photon collection efficiency in confocal. (c) Pinole size compared to PSF size.

** A very confusing point is “the optimal confocal pinhole diameter in our situations should be between 0.5 AU and 1 AU depending on the structural complexity” How can the value depend on the specimen?? The authors early stated that the resolution does not depend on the sample?*

Response: The selection of the optimal confocal pinhole diameter is influenced by the balance between photon collection efficiency and the effective Optical Transfer Function (OTF) enhancement. This balance is not uniform across all frequencies, which leads to varying optimal pinhole sizes depending on the specific spatial frequencies of the sinusoidal grating being imaged.

For example, at low frequency, confocal with a large pinhole collects more photons resulting in a clearer image. However, at higher frequencies, the benefit of collecting more photons with a larger pinhole is offset by a significantly reduced contrast due to the limitations in the OTF (Fig. R11a). For an object of specific frequency, there is an optimal pinhole that generates the largest information density (Fig. R11b). This is because this optimal pinhole size strikes the best balance between photon collection and OTF improvement for that frequency.

Figure R11. Variation of information density I_d in a confocal system with different circular pinhole sizes. (a) Comparisons of information density from a wide-field system and confocal systems with varying pinhole diameters at different target frequencies. (b) The relationship between pinhole diameter and information density I_d for objects at a specific frequency, with the red star denoting the peak I_d . (c) Noisy image visualizations of the two vertical dashed line in (a).

Our information density measure is based on imaging sinusoidal patterns as testing object. Therefore, it depends on the choice of testing sample. In our previous response letter, we considered ‘resolution is a property of the imaging system’ to distinguish our theoretical resolution measure from practical image-based resolution.

** Oversampling. I think here the authors make a real mistake in their reasoning “.. this understanding is crucial in explaining why oversampling can enhance Fisher information. This is due to the smaller pixel sizes allowing a greater passage of higher frequencies even beyond Nyquist sampling requirement.”*

There are no frequencies above the Nyquist sampling rate at the detector expect noise. At Nyquist the low pass filtering effect of the discretization/pixels is just so that it does not influence the reconstruction of the signal anymore because the signal is band limited by the optics to half the Nyquist rate. That is the whole point of Nyquist frequency of course.

Oversampling does not transfer more information. The system is hard band limited by the acceptance angle of the lens. If the measure produces a better value with oversampling, the authors must make a very, very strong case how this can happen.

Response: In our manuscript, we consider that “applying a pixel size even smaller than that required by Nyquist sampling is beneficial”. This benefit arises not from capturing frequencies above the Nyquist limit, which are not present in the signal after being band-limited by the optics, but rather from the way pixels integrate photons and the low pass filter created by this integration.

Camera pixels serve a dual role: they determine the sampling interval of the image, and they also integrate photons within their area, acting as a low pass filter. This filtering effect can be approximated by a sinc function, as detailed in **Fig. R13** and **Supplement Note 3**.

When forming an image with a camera, the process involves pixelating the underlying ideal image, which is frequency-filtered by the microscope's OTF. This pixelization can be conceptualized as binning the pixels of the underlying ideal image or, alternatively, as sampling the filtered image with the pixelization filter.

While by adhering to the Nyquist sampling theorem, one can recover the filtered image (as shown in Fig. R12d blue curve) from the camera ideal image (Fig. R12c), this recovered image does not fully represent the underlying ideal image. The reason is that some high-frequency

information is irretrievably lost due to the integration of pixel intensity.

Figure R12. Ideal image formation on the camera in a microscope system demo in 1D. (a) Object is generated from uniform distributed random number between [0,1]. It contains 200 underlying pixels covering 4 μm with each pixel size 0.02 μm . The microscope system is simulated with an NA of 1.25, a wavelength 0.5 μm . (b) Underlying ideal image is object frequency filtered by OTF. The camera pixel size is simulated 0.1 $\mu\text{m} = \frac{\lambda}{4NA}$ at the Nyquist sampling requirement. (c) Camera ideal image is generated by binning five pixels of underlying ideal image into one pixel and divided by five (scale for comparison purpose). Scaled camera ideal image equals the sampled value of filtered image (sample at every five pixel). (d) Filtered image is the underlying ideal image frequency filtered by the pixelization filter (Fig. R13).

Figure R13. OTF and low pass filter by pixelization. The low pass filter of pixelization is a sinc function $\frac{\sin(\pi kd)}{\pi kd}$ where d is the pixel size (Supplement Note3). When the pixel size is at the Nyquist sampling requirement: $d = \frac{\lambda}{4NA}$, the transmission rate at $k = \frac{2NA}{\lambda}$ is $\frac{\sin(\pi \frac{2NA}{\lambda} \frac{\lambda}{4NA})}{\pi \frac{2NA}{\lambda} \frac{\lambda}{4NA}} = \frac{2}{\pi} \approx 0.637$ (red curve).

While microscope system essentially performs a low pass filter on the image, pixelization (binning pixels) performs another layer of low pass filter on the image. The final image captured by the camera is thus a result of the image being filtered through these two sequential low-pass filters. The low-pass filter effect of pixelization is weaker compared to the OTF of the microscope system (Fig. R13). When pixel size is at the Nyquist sampling requirement: $d = \frac{\lambda}{4NA}$, the frequency transmission rate ranges from 0.637 at the diffraction limit frequency to 1 at the DC. Reducing the pixel size could improve the frequency transmission rate and increase the information density. Such improvement is obvious when the frequency is close to the diffraction limit boundary, while less pronounced when frequency is close to DC (**Fig. R14**).

Figure R14. Influence camera pixel size on information density on wide-field microscope. (a) Surface plot of information density versus sinusoidal grating's frequency and pixel size up to Nyquist requirement ($\frac{\lambda}{4NA} = 0.125\mu\text{m}$). (b) Plots of information density versus pixel size at low frequency. (c) Plots of information density versus pixel size at low frequency. In the above plots, the photon emission density was set as 5,000 photons/ μm^2 , with a numerical aperture of 1.4, an immersion medium refractive index of 1.5, and an emission wavelength of 0.7 μm . The photon collection efficiency is 32.05%, calculated from solid angle collection range.

Importantly, the pixelization frequency filter only affects the camera-based imaging technology where there is photon integration on the pixel. Scanning based imaging such as confocal would not be affected.

In the updated manuscript, we refined our manuscript to better elucidate the impact of pixelization in camera-based imaging systems, which involves photon integration on each pixel. We also revised the definition Nyquist sampling pixel size. Previously, it was defined as $\frac{1}{2k}$,

correlating to the reciprocal of twice the frequency of the tested sinusoidal grating object. Now, we have standardized this to a fixed value of $\frac{\lambda}{4NA}$, which corresponds to the reciprocal of twice the diffraction limit frequency.

Reference

1. Cox, G. & Sheppard, C. J. R. Practical limits of resolution in confocal and non-linear microscopy. *Microsc Res Tech* **63**, 18–22 (2004).
2. Wilson, T. The Role of the Pinhole in Confocal Imaging System BT - Handbook of Biological Confocal Microscopy. in (ed. Pawley, J. B.) 167–182 (Springer US, Boston, MA, 1995). doi:10.1007/978-1-4757-5348-6_11.

Reviewer #2 (Remarks to the Author):

The authors have satisfactorily addressed my concerns, and I am happy for this paper to be published.

As before, I am sure this paper will be well-received.

Reviewer #3 (Remarks to the Author):

The authors have taken away nearly all of my concerns. There remain only two that need some attention/clarification.

* emitted version detected photons:

Here the authors agree with me that the emitted photons are not relevant. I cannot understand that they still stick to the emission. The geometrical reasoning on the collection is limited to the objective alone. But even then that is OK if the emitter is far away from the interface, but in TIRF near field effects play a dominate role and super critical angle fluorescence will even outnumber the normal fluorescence calculated from your open angle consideration.

Over all I still think that fundamentally, the emitted number of photons is irrelevant but more importantly that for any practitioner in the field this number is inaccessible - deteriorating your resolution measure in use and acceptance in the field.

I do not understand why the emitted photon counts are needlessly used. This makes it also more difficult for the authors. The detected count is not arguable. The relation between emitted and collected photons based on the geometric considerations it not good. Now it matters eg. if the emitter how close the emitter is to the focal plane.

* I understand the point about object influence the pinhole size now. I do not see this reasoning reflected in the text. Or did I miss it.

* On the sampling and information loss/gain

+ I agree, the low-pass filtering effect due to non-ideal point sampling, but integration gives an additional blurring.

+ "Importantly, the pixelization frequency filter only affects the camera-based imaging technology where there is photon integration on the pixel. Scanning based imaging such as confocal would not be affected."

how can that be? Imaging you display a camera based and confocal with the same pixel size, you say that the confocal does not suffer from the same low-pass filtering due to pixelation? the point detector in confocal is also not a point but has a finite size, which you state in Airy units ?

+ I also see that the new figure 6 in revision 2 compared to revision 3 is changed a lot without commenting on it. The value on the y-axis is largely different. Now oversampling only increases the information density from 20 to 25 from Nyquist to large oversampling, where before this was 10 -> 25. A large difference.

In addition the x-axis changed, while the optical parameters of lambda, NA and photon count remained the same.

The diffraction limit for the given values is $700/2/1.4 = 250$ nm and therefore Nyquist is at 125 nm which seems about correct in the new figure, but is clearly off in the old figure.

Could the authors comment on this change?

Responses to Reviewers' Comments on

Manuscript: "A statistical resolution measure of fluorescence microscopy with finite photons"

Reviewer #2 (Remarks to the Author):

The authors have satisfactorily addressed my concerns, and I am happy for this paper to be published.

As before, I am sure this paper will be well-received.

Response: We thank the reviewer for the positive comments.

Reviewer #3 (Remarks to the Author):

The authors have taken away nearly all of my concerns. There remain only two that need some attention/clarification.

** emitted versus detected photons:*

Here the authors agree with me that the emitted photons are not relevant. I cannot understand that they still stick to the emission. The geometrical reasoning on the collection is limited to the objective alone. But even then that is OK if the emitter is far away from the interface, but in TIRF near field effects play a dominant role and supercritical angle fluorescence will even outnumber the normal fluorescence calculated from your open angle consideration.

Over all I still think that fundamentally, the emitted number of photons is irrelevant but more importantly that for any practitioner in the field this number is inaccessible - deteriorating your resolution measure in use and acceptance in the field.

I do not understand why the emitted photon counts are needlessly used. This makes it also more difficult for the authors. The detected count is not arguable. The relation between emitted and collected photons based on the geometric considerations is not good. Now it matters eg. if the emitter how close the emitter is to the focal plane.

Response: We agree with the reviewer that only detected photons form the image, which results in information density calculation. However, without referring to the emitted photon density, one will lose the capacity to use IbR to compare imaging modalities such as confocal and structured illumination microscope. By emphasizing the density of emitted photons, we aim to ensure an equitable comparison across different imaging techniques. We assume that the sample, when

labeled with fluorescent probes, exhibits consistent emission characteristics (such as the total number of emitted photons) irrespective of the imaging modality employed. This premise allows for a fair comparison of wide-field microscopy, confocal microscopy, and other modalities based on information density.

For readers interested in directly estimating the resolving power given a specific photon count detected, we invite them to examine the surface plot of information density versus frequency and the number of detected photons (**Fig. R1, also included in Supplementary Fig. 7**). We have now highlighted such supplementary data in our discussion in the main manuscript.

Figure R1. Fisher information calculation in four imaging modalities with respect to detected photons and frequency.

** I understand the point about object influence the pinhole size now. I do not see this reasoning reflected in the text. Or did I miss it.*

Response: We thank the reviewer for the suggestion. We have revised the manuscript, further explaining the influence of pinhole size on confocal systems.

* *On the sampling and information loss/gain*

+ *I agree, the low-pass filtering effect due to non-ideal point sampling, but integration gives an additional blurring.*

+ *“Importantly, the pixelization frequency filter only affects the camera-based imaging technology where there is photon integration on the pixel. Scanning based imaging such as confocal would not be affected.”*

how can that be? Imaging you display a camera based and confocal with the same pixel size, you say that the confocal does not suffer from the same low-pass filtering due to pixelation? the point detector in confocal is also not a point but has a finite size, which you state in Airy units ?

Response: Throughout our manuscript, we generally considered the equivalent pixel size of confocal to be the scanning interval. Our statement in the last response letter highlights that decreasing the pixel size in wide-field microscopy could enhance the information captured due to the reduced integration-based low-pass filter effect. Conversely, reducing the scanning interval in confocal microscopy does not generate a similar impact on the information density, as the integration-based low-pass filter effect is primarily influenced by the pinhole size and is already incorporated into the section investigating pinhole effects (Fig. R2).

Figure R2. Fisher information calculation in a wide-field microscope and a confocal microscope of different pixel sizes/scanning intervals. Confocal microscopes have pinhole sizes of 0.5 AU. Applying a pixel size smaller than the Nyquist sampling requirement increases information density in the wide-field microscope. Applying a scanning interval smaller than the Nyquist sampling requirement does not influence information density in the confocal microscope.

+ I also see that the new figure 6 in revision 2 compared to revision 3 is changed a lot without commenting on it. The value on the y-axis is largely different. Now oversampling only increases the information density from 20 to 25 from Nyquist to large oversampling, where before this was 10 -> 25. A large difference.

In addition the x-axis changed, while the optical parameters of lambda, NA and photon count remained the same.

The diffraction limit for the given values is $700/2/1.4 = 250$ nm and therefore Nyquist is at 125 nm which seems about correct in the new figure, but is clearly off in the old figure.

Could the authors comment on this change?

Response: In our first round submission, we considered the Nyquist sampling frequency as twice the frequency of the object, which is $k_{Nyquist} = 2k_{obj}$, where, in this specific case, the pixel size corresponds to $\frac{1}{2k_{obj}} = 0.2 \mu m$. In our second round of revision, as suggested by the reviewer, we changed the definition of Nyquist sampling frequency of microscope as twice the highest frequency the imaging system permits $k_{Nyquist} = \frac{4NA}{\lambda}$, where, in this specific case, the pixel size corresponds to $\frac{\lambda}{4NA} = 0.125 \mu m$.